# Elicitation of Novel Trichogenic-Lipid Nanoemulsion Signaling Resistance Against Pearl Millet Downy Mildew Disease

**DOI:** 10.3390/biom10010025

**Published:** 2019-12-23

**Authors:** Boregowda Nandini, Hariprasad Puttaswamy, Harischandra Sripathy Prakash, Shivakanthkumar Adhikari, Sudisha Jogaiah, Geetha Nagaraja

**Affiliations:** 1Department of Studies in Biotechnology, University of Mysore, Manasagangotri, Mysuru 560 006, Karnataka, India; bnandini2010@gmail.com (B.N.); geethabiotech.uom@gmail.com (H.S.P.); 2Centre for Rural Development and Technology, Indian Institute of Technology Delhi, New Delhi 110 016, India; phimprovement@rediffmail.com; 3Laboratory of Plant Healthcare and Diagnostics, PG Department of Biotechnology and Microbiology, Karnatak University, Dharwad 580 003, India; adhikariklr93@gmail.com

**Keywords:** nanoemulsion, *Sclerospora graminicola*, lipids, pearl millet, seed priming, *Trichoderma* spp.

## Abstract

Nanoemulsion was formulated from membrane lipids of *Trichoderma* spp. with the non-ionic surfactant Tween 80 by the ultrasonic emulsification method. Nanoemulsion with a droplet diameter of 5 to 51 nm was obtained. The possible effects of membrane lipid nanoemulsion on pearl millet (PM) seed growth parameters and elicitation of downy mildew (DM) disease resistance in PM was analyzed to develop an eco-friendly disease management strategy. Seed priming with nanoemulsion illustrates significant protection and elevated levels of early defense gene expression. Lipid profiling of *Trichoderma* spp. reveals the presence of oleic acid as a major fatty acid molecule. The prominent molecule in the purified lipid fraction of *T. brevicompactum* (UP-91) responsible for the elicitation of induction of systemic resistance in PM host against DM pathogen was predicted as (E)-N-(1, 3-dihydroxyoctadec-4-en-2yl) acetamide. The results suggest that protection offered by the novel nanoemulsion formulation is systemic in nature and durable and offers a newer sustainable approach to manage biotrophic oomycetous pathogen.

## 1. Introduction

Millets are referred to as “food medicine”. Millet is a source of antioxidants, such as phenolic acids and glycated flavonoids. Millet foods are also potential prebiotics and can improve the viability of probiotics with possible health benefits [1]. Pearl millet (*Pennisetum glaucum* (L.) R.Br. Syn. *Pennisetum americanum* (L.) Leeke) (PM) is the sixth most important cereal crop grown in the semi-arid and arid regions of the world [2]. In India, the crop is cultivated in a seven million ha area with a production of 9.25 million tons [2]. Pearl millet (PM) is the durable cereal and chief food source for people in drought-prone areas of Africa and India [3,4]. Pearl millet (PM) crop production is severely hampered by several biotic stresses. Downy mildew (DM) diseases caused by the oomycete obligate pathogen, *Sclerospora graminicola* (Sacc.) Schroet., is one of the major biotic constraints. The oomycetes vary from fungi and are incorporated amongst economically important plant pathogens, like DMs of Poaceae, Cucurbitaceae, and Vitaceae [5]. DM accounts for a yield loss of PM of up to 20% to 40% annually [6,7]. Spread of DM disease is favored by high relative humidity (85%–90%) with moderate temperatures (20–30 °C) [3,7]. With rising ecological attentiveness, the hub of managing plant diseases has been changed in the direction of employing feasible and sustainable alternative approaches [8].

*Trichoderma* is one of the most important and widely used fungal biocontrol agents. The biological activity of the species is related to the range and diverse metabolites that they produce [9]. Many biopesticides formulated from different *Trichoderma* spp. are available commercially [10]. A number of *Trichoderma* strains have been shown to colonize roots to elicit induce systemic resistance ISR, thus priming the host for an intense defense action against succeeding pathogen assault [11,12]. Various species of *Trichoderma* have been examined for their lipid production. Recently, a new Δ−10(E)-sphingolipid desaturase with heterologous expression was identified as a fundamental tool for biotechnological application and it has alleviated the relation of the immunosuppression and stress-tolerance mechanism in *Fusarium graminearum* [13].

Biocontrol organisms induces resistance in a host plant by elicitation of defense response through binding with a specific receptor signal molecule, which is carried by a compound known as elicitors. Elicitors, both biotic/abiotic, will bind to specific receptors and elicit the defense gene expression in host plants [14,15]. Ramsey et al. [16] demonstrated that surfactants with high ethoxylation will facilitate the permeability of water-soluble compounds and also improve the water content of the cuticle. Studies on *Trichoderma* and its membrane elicitors’ formulation helps us to understand the host–pathogen interactions and mechanisms involved in pathogen-associated molecular patterns (PAMPs). Small molecules like Tween 20 are more effective as a surfactant in reducing the droplet diameter than polymeric surfactants and are rapidly adsorbed onto the emulsion droplet surface [17]. An ethoxylated surfactant increases the diffusion of the sulfated laminarin through the leaf cuticle and stomata, resulting in enhanced induced resistance against grapevine DM [18].

Lipids from biological systems have gained special importance in crop disease management. Generally, microbial- or damage-associated molecular patterns (MAMPs/DAMPs) are associated with a number of lipid cascades. Nanoemulsion is one of the most accepted formulations because of its improved bioavailability, optical transparency, and greater physical stability [19,20]. Physico-chemical properties of nanoemulsions are influenced by several features, such as oil type, surfactant, and process conditions [21,22]. In the nanoemulsion process, densities, viscosities, and surface-active components of the different types of oils will considerably influence the emulsion quality in its composition of the dispersed oily phase [21]. Studies on *Trichoderma* and its membrane lipid nanoemulsion formulations help us to know the host–pathogen interactions and mechanisms involved in PAMPs. The objective of the present study was to formulate the nanoemulsion of a membrane lipid extracted from different *Trichoderma* spp. and to evaluate the efficacy of the synthesized nanoemulsion for the induction of disease resistance in PM against the DM pathogen.

## 2. Materials and Methods

### 2.1. Host, Pathogen, and Inoculum Preparation

The seeds of PM cultivar 7042S susceptible to DM were used as the host. Briefly, 10 downy mildew-infected young leaves from 7042S cultivar were collected in the evening and washed in running tap water to remove the remnants or old sporangia. Later, the leaves were cut into small pieces and petri dishes lined were with a wet double layer of blotter discs and the plates were incubated at 20 °C and 95% relative humidity (RH) in the dark overnight. In the early morning, sporangia produced on the leaves were harvested into sterile distilled water (SDW) and the spore load was adjusted to 4 × 10^4^ zoospores mL^−1^ using a hemocytometer, and used as a source of inoculum, according to Nandini et al. [23].

### 2.2. Trichoderma Spp.

Potent *Trichoderma* spp. (*T. asperellum* (DL-81: KM100835), *T. harzianum* (HR-73: KM100834), *T. atroviride* (MH-50: KM100830), *T. virens* (MP-60: KM100832), *T. longibrachiatum* (MP-59: KM100831) and *T. brevicompactum* (UP-91: KM100836)) used in the study were collected from the culture collection of the D.O.S in Biotechnology, University of Mysore. These fungi were originally isolated from the PM rhizosphere soil samples and were well characterized individually for their ability to promote plant growth and suppress DM disease (unpublished data). All the *Trichoderma* spp. were subcultured once in 15 days and maintained on potato dextrose agar (PDA) at 28 ± 1 °C throughout the experimental period.

### 2.3. Extraction of Total Lipids from the Trichoderma Mycelia

The selected six *Trichoderma* spp. viz., *T. asperellum* (DL-81: KM100835), *T. harzianum* (HR-73: KM100834), *T. atroviride* (MH-50: KM100830), *T. virens* (MP-60: KM100832), *T. longibrachiatum* (MP-59: KM100831), and *T. brevicompactum* (UP-91: KM100836), were mass-cultivated on potato dextrose broth for 12 to 14 days at 28 ± 2 °C. Towards the end of the incubation period, mycelia were harvested and blot-dried. Chloroform and methanol extraction was used to extract the lipid content from the fungal biomass. The wet biomass (5 g) was mixed with 15 mL. 

The solvent mixture of chloroform and methanol (2:1 *v*/*v*) was then incubated for 2 h in an agitator at 100 rpm. The mixture was then centrifuged at 10,000 rpm for 10 min. The solvent mixture was then separated into three different layers. The residual fungal biomass was in the bottom layer, the middle phase was the chloroform-containing lipid, and the top layer consisted of methanol and water. The middle phase of the chloroform-containing lipid was pipetted carefully and transferred into a pre-weighed glass tube (L_1_). The rest of the solution containing methanol and cell debris was again fortified with 15 mL of chloroform and methanol (1:2 *v*/*v*) solvent mixture and again incubated with agitation for 2 h. After incubation, the solution was centrifuged as explained above and the middle phase filtrate was mixed with previously extracted solution (chloroform solution containing lipid) and the mixed solution was allowed to stand for phase separation. The bottom phase of the chloroform-containing lipid (the upper phase was water and methanol) was collected and subjected to a vacuum rotary evaporator to concentrate the sample. Finally, nitrogen sparging was done until total chloroform got evaporated. The samples were further dried in an oven at 60 °C until a constant weight (L_2_) (Figure 1). The lipid recovery from the fungal biomass was calculated as:$$L\%=\frac{L_{2}-L_{1}}{DBW}\times 100$$

The obtained lipid was stored for further studies. In the above equation, CL represents the weight obtained from conventional lipid extraction, L_1_ expresses the pre-weighed glass tube, L_2_ denotes the oven dried microbial lipid in a pre-weighed glass tube, and DBW denotes the dry biomass weight.

The presence of lipids was confirmed by thin layer chromatography (TLC). The plates were developed with chloroform/methanol/water (65:25:4 *v*/*v*). Separation of lipids was visualized by exposing the TLC plates to iodine vapor, which is absorbed by both saturated and unsaturated lipids [24]. The developed plates were then sprayed with orcinol solution (20 mg orcinol in 10 mL of 70% H_2_SO_4_ (*w*/*v*)) and then heated at 100 °C for 10–15 min.

### 2.4. Preparation of Lipid Nanoemulsion and Its Stability Evaluation

Nanoemulsion was prepared by using 90% (*w*/*w*) of water, 5% (*w*/*w*) of lipid, and 5% (*w*/*w*) of polysorbate Tween 80 [25]. Both the lipid and surfactant were stirred at 800 rpm for 30 min using a magnetic stirrer. Subsequently, water was added dropwise at a flow rate of 3.5 mL/min. The mixture was stirred at 800 rpm for 60 min. Later, the solution was subjected to sonication at a high frequency of 20 kHz and power output of 750 W for 1 h. The resulted solution was used for further studies.

The nanoemulsion was stored at room temperature (20 ± 2 °C) and also at refrigerator temperature (4 °C). The stored samples were evaluated up to 60 days to check the active thermodynamic stability and also monitor for aggregation of the nanoemulsion during the period of storage. Stored samples were frequently observed for the formation of agglutination, creaming, and sedimentation of particles.

### 2.5. Characterization of the Lipid Nanoemulsion

Prepared lipid nanoemulsion was characterized using Nanotrac wave particle size analyzer SL-PS-25 Rev, in which the droplet size and its distribution, mean droplet diameter and polydispersity index (PDI), Zeta potential, and molecular weight can be analyzed. This instrument is capable of measuring from 0.8 to 6500 nm. The apparatus is equipped with a laser diode, 780 nm, 3 mW nominal, and temperature control from 5 to 90 °C using a Peltier Device. Dynamic light scattering is incorporated with the controlled reference method for advanced power spectrum analysis of Doppler shifts under Brownian motion and is capable of measuring up to a 40% concentration, which reduces the need to dilute samples. Turbidity analysis of the formulated emulsions was carried out by measuring the absorbance of undiluted samples at 600 nm (Hitachi U-3900 spectrophotometer, Tokyo, Japan).

### 2.6. Pearl Millet Seed Priming

Extracted total lipids of *Trichoderma* spp. were primed to the PM seeds in two different methods.

**Method I:** Extracted total lipids of six *Trichoderma* spp. were soaked in different concentrations of 50, 100, and 150 µg/mL in 20 mM potassium phosphate buffer (pH 6.5, containing 0.1% Tween 20, *v*/*v*) for 12 h at room temperature on a shaker at 150 rpm [26].

**Method II:** Sterilized seeds were coated uniformly with the lipid nanoemulsion for 12 h at room temperature on a shaker at 150 rpm separately. The seeds treated with SDW and 5% Tween 80 served as controls.

Seed priming with different concentrations of lipids and nanoemulsions was initially carried out at different time intervals of 3, 6, 9, and 12 h. Among the different time interval studies, 12 h of seed priming has shown significant influence on seed quality parameters compared to other time intervals. Hence, for the future experiments, only a 12-h incubation period of seed priming was considered.

### 2.7. Effect of Pearl Millet Seed Priming with Lipid Extracts on Seedling Germination and Vigor under Laboratory Conditions

A germination test was carried out by the paper towel method according to the standard procedures of International Seed Testing Association. The length of the root and shoot of individual seedlings was measured. The seedling vigor was analyzed as per the method of Abdul Baki and Anderson [27]. The vigor index (VI) was calculated using the formula:VI=(Mean root lenght+Mean shoot lenght)×percentage of germination

For each treatment, four samples of 100 seeds were used and the experiment was repeated thrice.

### 2.8. Anti-Mildew Activity

The leaf disc method was followed to assess the anti-mildew activity of the extracted elicitor against DM pathogen of PM. Diseased leaves were collected from the DM sick plot and washed with distilled water and blot-dried to remove excess water. The leaf discs of 10 mm in diameter were cut using cork borers and dipped in different concentrations of synthesized *Trichoderma*-mediated lipids and nanoemulsion (50, 100, 150 µg/mL) for 5 min. Treated and control leaf discs were placed in a dark moist chamber at 20 °C and 95% relative humidity (RH) for 12 to 16 h. Then, the treated and control leaf discs were placed (adaxial surface downwards) on moist blotter papers in petri dishes and the plates were incubated for 12 to 14 h in moist chambers (>70% RH and 20 °C) under dark condition. After incubation, the treated leaf discs were observed for sporulation under a stereo binocular microscope.

### 2.9. Zoosporicidal Assay

Fresh sporangia were harvested from the infected leaves as explained earlier. Immediately, the sporangial concentration was adjusted to 5 × 10^3^/mL using a haemocytometer and further used as inoculum. The reaction mixture (100 µL) of different concentrations of elicitors and 100 µL of inoculum was incubated in the dark for one hour. At the end of incubation, 20 µL of 2, 3, 5-triphenyltetrazolium chloride (TTC) solution was added and further incubated for 30 min. The sporangia were observed under a compound microscope for the red-colored insoluble formazan. The resultant mixture was centrifuged at 8000 rpm for 8 to 10 min and pellet was washed three times with sterile distilled water (SDW). The pellet was mixed with 1 mL of 95% ethanol and incubated in a water bath at 85 °C for 30 min. The mixture was centrifuged at 8000 rpm for 8 min, and then 200 µL of supernatant were transferred to a microtiter plate and read at 485 nm against respective controls. The percent inhibition of zoospore release was calculated by considering the color developed in the control as 100% viability. The distilled water treatment served as a negative control. Metalaxyl was the chemical used as the positive control. The assay was performed in triplicates for each treatment:
$$\%\;Zoosporicidal\;activity=\frac{Absorbance\;of\;control-Absorbance\;of\;treatment}{Absorbance\;of\;control}\times 100$$


### 2.10. Effect of Seed Priming with Lipid Extracts on Pearl Millet—Downy Mildew Disease under Greenhouse Conditions

Seed priming was carried out as explained previously. The seeds treated with a systemic fungicide, Metalaxyl, at a 6 g/kg dose as the positive control. Sterile distilled water-treated seeds served as the negative control.

Control and treated seeds were sown in earthen pots (20 cm diameter) containing sterilized potting mixture (soil:sand:farmyard manure (FYM), 2:1:1). Each treatment consisted of eight replications of five pots with eight seedlings each and repeated thrice. The experiment was laid out as a randomized complete block design. Two-day-old seedlings were challenge-inoculated by the whorl inoculation method with the zoospore suspension of *S. graminicola* at a concentration of 4 × 10^4^ zoospores mL^−1^. In the whorl inoculation, droplets of *S. graminicola* zoospores were dropped onto the leaf whorl formed by the emerging seedlings and allowed to flow down to the base. The challenge-inoculated plants were maintained under greenhouse conditions (90%–95% RH, 20–25 °C temperature). Disease incidence was monitored by counting the number of plants that showed any one of the typical DM symptoms, which consisted of sporulation on the abaxial leaf surface, chlorosis, stunted growth, or malformation of the panicles. Disease incidence was recorded at 15, 30, and 45 days after challenge inoculation. The experiment was concluded 60 days after sowing.

### 2.11. Evaluation of Lipid Nanoemulsion of Trichoderma spp. against S. graminicola Infection in Pearl Millet

#### 2.11.1. Hypersensitive Reaction (HR)

The hypersensitive reaction (HR) in different time intervals was studied by following the procedure of Kumudini et al. [28]. Treated and control seeds were germinated in petri plates lined with wet blotter sheets (25 seeds/plate) for three days. Two-day-old seedlings were root-dip inoculated with a zoospore suspension of 4 × 10^4^ mL^−1^ and incubated in the dark at 25 (±1) °C [29]. PM seedlings were observed at 0-h intervals up to 24 h for the external appearance of necrotic spots or streaks on the coleoptile and/or root regions were observed under a stereo binocular microscope. PM seedlings subjected to different treatments as described earlier were observed at 0, 1, 2, 4, 6, 8, 10, 12, and 24 h for the external appearance of necrotic spots or streaks on the coleoptiles and/or root region of the test seedlings. The initial time of appearance of HR and the number of seedlings showing the necrotic spots until the experimental period of 24 h were recorded and the percentage was calculated. The experiment was repeated three times with 100 seedlings of four replications for each experiment:
$$\%\;HR=\;\frac{Number\;of\;seedlings\;with\;necrotic\;spots}{Total\;number\;of\;seedlings\;taken}\;\times 100$$


#### 2.11.2. Histological Studies

Lignification in the cell wall of the control and treated seedlings was visualized as explained by Sherwood and Vance [30]. The percentage of lignified cells was counted in 10 randomly selected microscopic fields and the average was tabulated. The experiment was repeated thrice, consisting of three replicates of 10 seedlings in each treatment observed over 20 microscopic fields.

Hydrogen peroxide (H_2_O_2_) deposition was studied following the method of Thordal-Christensen et al. [31]. The peelings were observed under a microscope for H_2_O_2_ staining. The cells with brown-colored staining deposition were counted, and the percentage was calculated. The experiment was repeated thrice, consisting of three replicates of 10 seedlings observed over 20 microscopic fields.

Callose deposition studies were carried out following the procedure of Jensen [32]. The cells with callose-deposition fluoresced and the callose-deposited cells were counted and the percent cells with callose deposition were calculated. The experiment was repeated thrice, consisting of three replicates of 10 seedlings each observed over 20 microscopic fields.

#### 2.11.3. Durability of Resistance

Susceptible PM cv. 7042S were treated with a lipid nanoemulsion of *T. brevicompactum* and sown in earthen pots. The plants were maintained under greenhouse conditions. The treatments were positioned in a completely randomized block design. Seeds treated with sterile distilled water and Tween 80 served as the control. One set of plants was challenge inoculated with pathogen at the tillering stage and another at the emergence of the boot leaf stage with the DM pathogen. The emerging leaf whorls of the basal tillers (25–30-day-old plants) and the inflorescence axis at the boot leaf stage (45-day-old plants) received the second dose of inoculum. The effect of the membrane lipid nanoemulsion formulation was compared with that of the systemic fungicide, metalaxyl. Each treatment was replicated four times with 20 pots per replication and 20 seedlings per pot.

#### 2.11.4. Field Studies

Lipid nanoemulsions of the *T. brevicompactum* treatment offering the best protection under greenhouse conditions against DM disease were selected for field trials. Field studies were conducted in the DM sick plot situated in DOS in Biotechnology, University of Mysore (N24°18′, E79°26′, 903 m altitude, red loam soil). This experimental station has been maintained over three decades under the Indian Council of Agricultural Research (ICAR) program, with severe infestation of the DM oospores inoculum. Additional inoculum in the form of asexual spores is provided from spreader rows raised 21 days prior to sowing of the experimental plot [3]. Seed priming was done as described above. The treated seeds were sown in a randomized block design of 10 × 6 m plots with four replications per treatment. Each row was 6 m long and 75 cm apart with 15-cm spacings between plants within rows. Susceptible controls were similar to that described earlier. The plants were raised following recommended agronomical practices. Disease incidence was recorded at 30- and 60-days of growth of seedlings, as the plants started showing typical DM disease symptoms.

### 2.12. Characterization of Fatty Acids

#### 2.12.1. Fatty Acid Extraction and Fatty Acid Methyl Esters (FAMEs) Preparation

The fungal lipid was converted to volatile fatty acid methyl esters (FAMEs) derivatives by following the method of Saini et al. [33]. Separations were done to convert lipid to volatile derivatives by the conventional anhydrous methanolic HCl method.

FAME were analyzed by GC–MS (PerkinElmer, Turbomass Gold, Mass spectrometer) equipped with a flame ionization detector (FID) using a fused silica Rtx-2330 column (Restek made, 30 m × 0.25 mm ID, and 0.25 µm film thickness). The injector port and detector temperatures were set at 230 and 250 °C, respectively, and N2 was used as the carrier gas. Initially, the column temperature was maintained at 120 °C, followed by increasing it to 220 °C over 20 min, and holding it at 220 °C for 10 min. The FAMEs were identified by comparing their fragmentation patterns and retention times with authentic standards and also with the National Institute of Standards and Technology (NIST) library.

#### 2.12.2. Purification of Total Lipid Extract of *T. brevicompactum*

Extracted membrane lipids from the *T. brevicompactum* were purified by column chromatography in which neutral lipids, glycolipids, and phospholipids were recovered by elution with chloroform, acetone, and methanol. The acetone and methanol fractions eluted the sphingolipid components of the sample and were further purified on another silica gel column with sequential elution with chloroform/methanol with an increasing concentration of methanol (95:5, 9:1, 8:2 *v*/*v*) and finally with 50% methanol. Sphingolipid fractions eluted with chloroform and methanol (9:1 to 8:2) were collected and further purified by chromatography on latrobeads RS 2060 (Duren, Germany) using the same elution system as prescribed above, resulting in accomplishing the purified sphingolipid fraction of *T. brevicompactum* lipid.

#### 2.12.3. Thin Layer Chromatography (TLC) Analysis

Each fraction collected during the purification process was examined to detect the presence of lipids. The TLC plates (0.2 mm thick silica gel, Merck, Kenilworth, USA) were developed and the presence of lipids was visualized.

#### 2.12.4. Sugar Analysis

Thin layer chromatography (TLC) analysis of purified fractions of *T. brevicompatum* indicated the presence of sugars in the purified fractions with the appearance of pink-violet on a white background. The presence of sugars in the purified fraction was detected by Molish’s test. The fraction was further subjected to TLC preloaded with 0.2 mm thick silica gel by a double ascending method in a solvent system consisting of 1-butanol, 2-propanol, and water (1:7:2) using aniline phthalate (0.93 g of aniline and 1.66 g of α-phthalic acid dissolved in 100 mL of n-butanol saturated with water) as a spraying reagent for the visualization of sugars in TLC plates.

#### 2.12.5. Liquid Chromatography-Mass Spectroscopy (LC-MS) Analysis

Purified lipid fraction from the *T. brevicompactum* was analyzed by LC-MS to identify the structural information. Liquid chromatography-mass spectroscopy (LC-MS) measurements were performed using Water’s Synapt G2 (UHPLC-ESI-QTOF) equipped with an automatic injector, a column thermostat, and the electrospray ionization source, the Mass Lynx data processing system. Separation was carried out on an Aquity BEH C18 (50 × 2.1 mm, × 2.7 µm). Gradient elution (elution of 5% of B in 0.0 min, 95% of B in 3 min, 95% of B in 5 min, 5% of B in 6.5 min, and 5% of B in 8.0 min) was applied with a mobile phase of 0.1% formic acid (mobile phase A) and acetonitrile (mobile phase B at the flow rate of 0.3 mL min^−1^. The column temperature was maintained at 45 °C and the injection volume of the sample was 10 μL. The LC-MS was operated in the positive mode with optimized condition: Desolution gas 350 L/h, source temperature 250 °C, capillary voltage 2300 V, detection range 50–1800 m/z.

#### 2.12.6. Fourier Transform Infrared Spectroscopy (FTIR) Analysis

The assignment of functional groups in the purified lipid fraction of *T. brevicompactum* was analyzed by FTIR spectrum (PerkinElmer Spectrum NIOS2) with KBr. The spectrum was recorded at a resolution of 4 cm^−1^ with an MIR TGS detector in the range 500–4000 cm^−1^.

#### 2.12.7. Nuclear Magnetic Resonance (NMR) Analysis

The purified sphingolipid sample was dissolved in 0.7 mL of deuterated choloroform (CDCl_3_) and used for NMR analysis. Proton (^1^H) and carbon 13 (^13^C) NMR analysis was performed for the purified sphingolipid sample using one-(1D) and two-dimensional (2D) NMR gradient-selected correlation spectroscopy (gCOSY) and nuclear overhauser effect spectroscopy (NOESY) techniques. NMR spectra were recorded on an Agilent 400-MHZ spectrometer operated in the Fourier transform mode. The ^1^H NMR experiments were reported in δ units, parts per million (ppm), and were measured relative to the residual chloroform (7.26 ppm) in the deuterated solvent. The ^13^C NMR spectra were reported in ppm relative to deuterochloroform (77.0 ppm). All hydrogen-coupling constants were examined as illustrative of all the hydrogen signals’ multiplicities.

NMR spectra were assigned through gradient-correlation spectra (g-cosy) and distortionless enhancement by polarization transfer (DEPT-135). The assignment of hydrogenated and non-hydrogenated carbon was carried out by accompanying the ^13^C and DEPT-135. All coupling constants, *J*, were reported in Hz. The following abbreviations were used to describe the peak splitting patterns when appropriate: s = singlet, d = doublet, t = triplet, dd = doublet of doublet, and m = multiplet.

### 2.13. Modulation in Defense Enzyme Activities after Seed Priming of Pearl Millet Seeds with T. brevicompactum Lipid Nanoemulsion

Three-day-old seedlings grown on wet blotter discs in petri plates (25 seeds/plate) from treated and control seeds that were root-dip inoculated with a zoospore suspension of 4 × 10^4^ mL^−1^ and incubated in the dark at 25 (±1) °C. Seedlings (1 g fresh weight) were harvested at 0, 3, 6, 12, 24, 36, 48, 72, and 96 h after challenge inoculation, ground to a fine powder in liquid nitrogen, and used for extraction. The protein content of the extract was estimated using the dye binding method of Bradford [34] with bovine serum albumin (Sigma, St. Louis, USA) as the standard.

Lipoxygenase (LOX) Assay (EC 1.13.11.12)

Lipoxygenase activity was measured by following the procedure of Borthakur et al. (1987). Supernatant of 0.5 g of seedlings extract with 5 mL of 0.2 M sodium phosphate buffer (pH 6.5) was used as the enzyme source. The activity was determined spectrophotometrically by monitoring the appearance of the conjugated diene hydroperoxide at 234 nm. The substrate for LOX assay was prepared according to the method described by Pushpalatha et al. [35]. The experiment was performed thrice and the means of the enzyme activity were tabulated.

Allene Oxide Synthase (AOS) Assay (EC 4.2.1.92)

Enzymes were extracted from all the harvested samples according to the method reported by Pushpalatha et al. [36].

Preparation of Fatty Acid Hydroperoxide

13-hydroperoxides of linolenic acid (13-HPOT) used as the substrate for the AOS assay was preparedusing soybean lipoxygenase and linolenic acid. The AOS 13-HPOT-metabolizing activity in various PM seedlings was measured spectrophotometrically by monitoring the rate of the decrease in absorbance at 234 nm [37]. The enzymatic assays were performed at 25 °C in 1 mL of 50 mM phosphate buffer, pH 7 containing 20 µL of 13S-hydroperoxylinolenic acid (stock solution: 1 mg/mL in ethanol) equal to an absorbance of 1.5 at 234 nm. The absorbance was recorded for 2 min following the addition of 50 µg of total protein (Hitachi U-3900, Tokyo, Japan) on a UV–visible spectrophotometer. The specific activity was expressed as a change in the absorbance at 234 nm/mg protein/min.

#### 2.13.1. Transcriptional Analysis

##### Total RNA Extraction and cDNA Synthesis

Total RNA was extracted from PM seedlings harvested at 72 h.a.i. using Trizol Reagent (Thermo Scientific, Bangalore). Extracted RNA was then treated with DNase I (RNase free) (Thermo Scientific, Bangalore) to eliminate contamination of the entire genomic DNA and stored at –80 °C. The quality of the RNA sample was evaluated on a 1.2% (*w*/*v*) agarose gel. The quantity and quality of RNA samples were also verified using a NanoDrop 2000 spectrophotometer (Thermo Scientific, Bangalore). cDNA was synthesized in a 25-μL reaction mixture containing 2 μg of RNA and 0.5 μg of oligo (dT)18 primer at 42 °C with 0.2 unit of RevertAid M-MuLV Reverse Transcriptase (Thermo Scientific, Bangalore, India) and 20 units of RiboLockRNase Inhibitor for 1 h.

##### Sequence and Phylogenetic Analysis

The conserved motifs of the lipoxygenase (LOX), allene oxide cyclase (AOC), and α-dioxygenase (α-DOX) sequence were analyzed at NCBI and the domains were predicted. Monocot gene coding sequences *Zea mays*, *Setaria italica*, and *Sorghum bicolor* for specific primers were obtained from the NCBI nucleotide database and aligned using the multalin online software. The evolutionary relationships of the nucleotide sequence from other plant species obtained from NCBI were examined by phylogenetic analysis using the maximum-likelihood approach at http://www.phylogeny.fr. The consensus sequences thus obtained were used to design the primer sets. The optimum annealing temperature was determined by gradient PCR for each set of primers in a separate experiment. The PCR amplification were confirmed on 1.2% agarose gel by submerged agarose gel electrophoresis against a GeneRuler™ 1 kb DNA Ladder (Thermo Scientific, Bangalore) under standard conditions.

##### Gene Expression Analysis through Quantitative Real-Time PCR (qRT-PCR)

The expression profile of LOX, AOC, and α-DOX in PM during DM pathogen infection was examined by the qRT-PCR technique. Details of the reference genes and target genes are summarized in Appendix A. qRT-PCR was performed in Step One Plus TM Real-Time PCR Systems (Applied Biosystems, Bangalore). An internal reference gene, like glyceraldeyde-3-phosphate dehydrogenase GAPDH (GQ398107), was used for normalizing and measuring the gene expression pattern. Fluorescence acquisition was achieved at 60 °C. A total reaction volume of 20 µL was prepared containing 10 µL of 1X SYBR Green PCR master mix reagent (Sigma-aldrich, Bangalore), 1 µL of 20 ng each of cDNA, and 3 pmol of 3 µL forward and reverse primers. The qRT-PCR cycling program was with initial denaturation at 95 °C for 10 min followed by 40 cycles of denaturation at 95 °C for 15 s, annealing, and extension for 60 s at 60 °C. A melting curve was generated at the end of each reaction, using a cycle consisting of 15 s of denaturation at 95 °C and 60 s of annealing at 60 °C followed by a slow temperature increase to 95 °C at the rate of 0.3 °C s^−1^. The comparative C_T_ (2^−ΔΔCT^) method [9,12] was used for evaluating the relative gene expression. All experiments for each data set for qRT-PCR were carried out with three replicates.

#### 2.13.2. Jasmonic Acid (JA) and Methyl Jasmonate (MeJA) Quantification in *T. brevicompactum* Nanoemulsion-Treated Seedlings by GC-MS Analysis

Estimation of the sensitivity of gas chromatography (GC) was the first step in the evaluation of the method. Chemical ionization in the single ion mode revealed the highest sensitivity for JA and MeJA. By using methylated standards, retention times were estimated, and a method was established to collect only relevant ions in a specific time [9]. Standards of JA and MeJA were obtained from Sigma-Aldrich. Quantification of JA and MeJA levels in PM seedlings was estimated for the seedlings collected at 48 h after pathogen inoculation, because at this particular time interval, significant upregulation of defense modulation of enzymes was observed. After establishing the GC program, the sensitivity of the GC was evaluated by using dilutions of known amounts of MeJA and JA in dichloromethane. Standard curves were measured twice, with four replicates for each added amount of compound. The estimation of the recovery rate revealed that JA and its corresponding internal standard exhibited the same properties as that expressed in the recovery rate. With the plant material serving as a matrix, the standard curve was linear over the range 0–1000 ng of added compound for JA and MeJA, with a correlation coefficient of 0.999.

Briefly, the frozen plant material (0.3–0.5 g of fresh weight) was pulverized with liquid nitrogen and sand using a pestle and mortar. An aliquot of methanol (1:2, *w*/*v*) was added, and the mixture was centrifuged at 7500 *g* for 15 min at 4 °C. The supernatant was collected in a glass vial, and the solvent was evaporated under liquid or vapor N_2_ at ambient temperature. Two milliliters of ethereal diazomethane were added to the dried sample for the derivatization reaction, and after 30 min, the reaction was stopped under a gentle N_2_ stream. One milliliter of 30% (*w*/*v*) NaCl solution was added to the dried sample in a vial sealed with a silicon septum containing a stir bar. Extraction of methyl jasmonate was carried out by headspace exposure of the polydimethylsiloxane-solid phase micro extraction (PDMS-SPME) fiber over the aqueous sample, stirring at 60 °C for 30 min. Blank analyses were carried out by exposure of the fiber to the saline solution.

Analyses were performed with a Varian CP-3800 gas chromatograph (Varian Inc., Palo Alto, California, USA) equipped with a 1177 split/splitless injector, a 30 m × 0.25 mm, i.e., 0.25 µm, CP-Sil8CB capillary column (Varian), an FID detector, and Galaxie Workstation software (Varian Inc. Palo Alto, California, USA). Desorption of the PDMS-SPME fiber was made directly into the injector port for 5 min at 250 °C in splitless mode. The injector split/splitless program mode was: 0–5 min splitless; 5.01–5.75 min at a 1:10 split ratio. The column oven was programmed as follows: 60 °C (1 min) to 280 °C (2 min) at 25 °C/min. The temperatures of the injector port and detector were 250 and 280 °C, respectively. Helium was used as the carrier gas and its pressure was maintained constant at 10.0 psi (1 mL/min). The experiment was performed thrice, and the mean JA and MeJA level was calculated [9].

### 2.14. Statistical Analysis

The data were analyzed separately for each experiment and subjected to analysis of variance using (ANOVA) SPSS Inc. version 17.0. A significant effect of treatments was determined by the F value (*p* ≤ 0.05). The significant differences between the treatment means were compared using the highest significant difference (HSD) as obtained by Tukey test at *p ≤* 0.05 levels.

## 3. Results

### 3.1. Characterization of Lipid Emulsion of Trichoderma spp. and Its Stability during Storage

Emulsification of crude lipids (5%) with non-ionic surfactant Tween 80 (5%) will help in the solubility of the extracted lipids in O/W formulation. Emulsified oil was sonicated for 1 h for the homogenization of droplets in the emulsion (Figure 2). Tween 80 surfactant with hydrophilic–lipophilic balance (HLB) leads to the formation of emulsified solution with a droplet size in the range 5–51 nm, in which *T. longibrachiatum* membrane lipid samples show a minimum droplet size of 5.94 nm with pH 7 and viscosity of 0.8110 with positive polarity. Table 1 summarizes the physico-chemical parameters of the membrane lipid nanoemulsion of all six different *Trichoderma* spp. in which the droplet size of the membrane lipid nanoemulsion with a balanced pH 7 is in all samples.

The stability of the emulsion was evaluated by storing the emulsified oil samples with and without sonication for a period of 15, 30, and 60 days in a room temperature of 25 ± 2 °C and at 4 °C in darkness. Initially, sonication was performed for 30 min and observed for its stability and homogenization under room temperature, but in that, settling of the particles was observed after a few days. A further sonication process was increased for 1 h and this sonicated emulsion was observed to be stable for 40 days, at room temperature and 60 days at 4 °C without agglutination, creaming, and any particle settlement at the bottom (Figure 3).

By visual observation, lipid emulsions showed sedimentation in the samples without sonication, whereas samples with sonication were stable for a long period. During storage conditions at room temperature, there were minor changes in the solution turbidity and aggregation of the droplets was observed in the sonicated emulsified sample after 40 days of incubation. The samples with sonication stored at 4 °C did not show any change in the emulsion samples until 60 days of incubation (Figure 3). In case of emulsified lipid samples without sonication and stored at room temperature, the particles sedimented within 12 h of incubation. Similar results were observed with the same samples, without sonication stored at 4 °C in darkness. During storage conditions at 4 °C, no notable visual change was observed in the emulsion samples with sonication.

### 3.2. Effect of Seed Priming with Lipid Extracts of Trichoderma spp. on Seed Quality Parameters of Pearl Millet

**Method I:** Seed priming with total lipid extracts of *Trichoderma* spp. in different concentrations had no effect on seedling quality. Seed priming with lipid extracts was found to improve the seedling vigor significantly (*p ≤* 0.05) at a concentration of 100 µg/mL; however, not much difference was observed in enhancing the seed germination in treated seedlings. Between the different treatments of six different *Trichoderma* spp. in different concentrations, *T. brevicompactum* at 100 µg/mL concentration showed the highest seedling vigor of 1623 with 92% germination compared to all other treatments. Potassium phosphate buffer (20 mM) pH 6.5, containing 0.1% Tween 20 showed 89% germination and vigor of 1552, which was not significantly different from the sterile distilled water control treatment. The metalaxyl seed treatment showed 89% seed germination and 1547 seedling vigor (Table 2).

**Method II:** Nanoemulsion seed treatment with *Trichoderma* lipid increased the seed germination and seedling vigor compared to the control treatments (Table 2). The lipid nanoemulsion treatment of *T. brevicompactum* and *T. asperellum* showed the maximum germination of 93% with seedling vigor of 1921 and 1879. Seedling vigor in treated seedlings significantly (*p ≤* 0.05) increased compared to the control treatment. The lipid nanoemulsion treatment of all six different *Trichoderma* spp. showed very good improvement in the seed quality parameters of PM.

### 3.3. Anti-Mildew Activity and Zoosporicidal Assay

The total lipid treatments in two different methods did not show sporulation inhibition in anti-mildew activity. However, the positive control, metalaxyl (0.6%), offered 93% inhibition of sporangia.

All tested *Trichoderma*-mediated lipid extracts in different concentrations of six different *Trichoderma* spp. were not able to inhibit zoospore release and arrest the motility of zoospores. In contrast, in the metalaxyl (0.6%) control treatment, only 3% zoospore release and 100% arrest of zoospore motility was observed.

### 3.4. Effect of Trichoderma Membrane Lipid Induced DM Disease Resistance in Pearl Millet under Greenhouse Conditions

**Method I:** Seed priming with total lipid extracts of six *Trichoderma* spp. at different concentrations of 50, 100, and 150 µg/mL in 20 mM potassium phosphate buffer (pH 6.5, containing 0.1% Tween 20, *v*/*v*) showed varied disease protection depending on the concentration. A significant (*p ≤* 0.05) protection of 68.6% was observed in the seedlings treated with *T. brevicompactum* at the 100 µg/mL concentration followed by *T. virens* with 63.9% protection. Potassium phosphate buffer at a concentration of 20 mM, pH 6.5, containing 0.1% Tween 20, *v*/*v* Tween 80 in a concentration of 5% showed any protection. No significant differences were observed between the disease protection mediated by *T. brevicompactum* and the positive control. Among the treatments, *T. longibrachiatum* showed the least efficacy to induce disease resistance. However, the metalaxyl positive control treatment gave the highest disease protection of 89.5% with the least disease incidence compared to the other treatments (Table 3, Figure 4).

**Method II:** Lipid nanoemulsion of *Trichoderma* spp. Induced resistance in PM against DM disease under greenhouse conditions, in which *T. brevicompatum* membrane lipid nanoemulsion treatment showed the maximum significant protection, with 84.8% protection, followed by *T. virens* with 70.3% protection. The lipid nanoemulsion treatment of *T. atroviride* and *T. asperellum* showed 64.4% and 58.2% protection. Control treatments like SDW and Tween 80 (5%) did not show any protection. Among the different treatments, the chemical positive control treatment offered the highest disease protection of 89.5% with the least disease incidence compared to other treatments.

Seed priming with total lipid extract of *Trichoderma* spp. in two different methods was done as explained above, in which seed priming of the lipid extract in the method II offered significant protection under greenhouse conditions compared to the method I approach and also method II enhanced the seed quality parameters significantly compared to the method I approach. So, based on these obtained results, we further continued the work with seed priming following the method II to identify the durability and modulation defense enzymes in pearl–DM interactions.

### 3.5. Evaluation of Hypersensitive Reaction in Nanoemulsion of Trichoderma Membrane Lipid Treated Pearl Millet Seedlings

The effect of the nanoemulsion of *Trichoderma* membrane lipids was evaluated based on the expression of the HR-like lesion response, i.e., brown necrotic spots/streaks in PM-treated seedlings upon root dip inoculation of DM pathogen at different time intervals. In *T. brevicompactum*, an HR-like lesion was observed as early as 3 h, and at 12-h time intervals, 6% and 27% HR-like lesion was recorded. Maximum HR-like lesion was recorded in treated seedlings at 24 h after pathogen inoculation (h.a.i.). Whereas in SDW control seedlings, the HR-like lesion response was delayed, and it was observed at 8 h.a.i. and the maximum was observed at 24 h.a.i with only 7% of HR-like lesion observed. Among the treatments, the Tween 80 (5%) treatment also showed the least HR-like lesion response and it was not significantly (*p ≤* 0.05) different from the control (Figure 5).

A constitutive level of lignification was observed in all treatments, i.e., both inoculated and seedlings, in the coleoptiles and root regions. However, after pathogen inoculation, treated seedlings showed lignification with a higher intensity as early as within 8 h of post inoculation. Nanoemulsion treatment of *T. brevicompactum* showed the highest lignin deposition of 46% and 62% at 12 h.a.i and 24 h.a.i. Whereas, in the case of *T. asperellum-*, *T. virens-*, and *T. atroviride-*treated seedlings, a moderate amount of lignification was recorded at 8 h and the maximum lignification was observed at 24 h.a.i. as 49% lignification was observed. Callose deposition was observed under epifluorescence microscopy, in which control seedlings has the least callose deposition of 3% at 24 h.a.i. Whereas in the case of the *T. brevicompatum* membrane lipid nanoemulsion treatment, seedlings showed a maximum deposition of 76% at 24 h.a.i. (Figure 6).

It was examined that deposition of H_2_O_2_ varied in the control, susceptible and elicitor treatments induced resistance response in the seedlings and sequentially increased with time after pathogen inoculation, and maximum deposition was noticed in *T. brevicompactum* nanoemulsion-treated seedlings that showed significant accumulation of H_2_O_2_ maximum at 24 h.a.i. with 89%. Nanoemulsions of *T. asperellum-*, *T. atroviride-*, and *T. virens*-treated inoculated seedlings showed 57%, 63%, and 69% H_2_O_2_ deposition. Control treatments of SDW and 5% Tween 80-treated seedlings showed the least deposition of 13% at 24 h.a.i. (Figure 6).

Callose deposition was observed under the fluorescence microscope and showed *β*-1, 3-glucan deposited between the cell wall and the plasma membrane. Callose deposition appeared as bright fluorescent irregular structures, sometimes surrounded by a halo region in the outer epidermal cell wall just beneath the appresoria. Fungal mycelium growth beyond this structure was hardly observed. Callose deposition was observed as maximum in *T. brevicompactum* nanoemulsion-treated seedlings, with 64% of deposition at 24 h.a.i, and was initially observed as early as 6 h.a.i. with 13%. Whereas *T. asperellum-*, *T. virens-*, and *T. atroviride*-treated seedlings showed deposition at 8 h.a.i. and the maximum was observed with 53% at 24 h.a.i. However, SDW control and 5% Tween 80-treated seedlings showed the least deposition of 5% at 24 h.a.i. compared to all other treated seedlings (Figure 6).

### 3.6. Durability of Induced Resistance by Nanoemulsion Treatment of T. brevicompactum Membrane Lipid in Pearl Millet Seedlings against Downy Mildew Disease

The durability of resistance was studied to know the disease development even after they were challenge inoculated at the tillering stage. When the plants were again challenge inoculated with the pathogen in its tillering stage, a significant (*p ≤* 0.05) protection of 83.7% was recorded. The same treatment yet again was re-challenged at its inflorescence stage, in which the degree of disease protection was observed in the same level with 82.5% protection. The positive control treatment of metalaxyl fungicide (Apron 35 SD) recorded 89% disease protection (Figure 7). However, the control treatment showed the highest disease incidence of 94% and 88% recorded at the tillering and inflorescence stage, respectively.

### 3.7. Field Studies

The effect of seed priming with *T. brevicompactum* lipid nanoemulsion formulation was evaluated in the DM sick plot under epiphytotic field conditions. Maximum protection with the least incidence was observed in *T. brevicompactum*-treated seedlings with 82.8% protection with the least disease incidence of 16.9%, but in the case of the control-treated plot, the highest disease incidence was noticed with 94% disease incidence and hence not offering any protection against DM pathogen in the host (Figure 8 and Figure 9). The seed quality parameters of the *T. brevicompactum* lipid nanoemulsion on the growth parameters of PM plants under field conditions are summarized in Table 4. Chemical metalaxyl treatment exhibited maximum protection with 89% disease protection.

### 3.8. GC-MS Analysis of FAME of Trichoderma spp.

In the present study, nine fatty acids were identified by GC–MS analysis of six *Trichoderma* spp. (Table 5). In all the studied *Trichoderma* spp., oleic acid (C18:1) in both the -cis and -trans position was found in the highest quantity except in *T. harzianum*. *T. virens* showed the highest quantity of -cis, oleic acid of 28.83% and -trans oleic acid was found in the highest quantity of 35.06% in *T. longibrachiatum*. Oleic acid was found in the least quantity in *T. harzianum* in both the -cis and -trans position in the range of 5.45% and 8.03%. Lauroleic acid (C12:1), myristic acid (C14:0), pentadecylic acid (C15:0), and palmitic acid (C16:0) were found in the range of 0.53% (*T. asperellum*) to 6.68% (*T. brevicompactum*). Palmitoleic acid (C16:1) was quiet variable among the species, in which *T. harzianum* showed the highest quantity with 35.07% and *T. virens* showed the least quantity with 0.75%. Stearic acid (C18:0) was found in a moderate quantity in all the *Trichoderma* spp. in the range of 19.72% (*T. brevicompactum*) to 28.45% (*T. virens*). The highest quantity of linoleic acid (ω6, C18:2) was recorded in *T. harzianum* with 7.24%. Arachidonic acid (C20:2) was found in the highest quantity in *T. virens* with 11.31%. Total polyunsaturated fatty acids and total monounsaturated fatty acids (PUFAs and MUFAs) were found in the ratio of 0.02 to 0.22. The total lipid content was recorded in the range of 3.23% (*T. asperellum*) to 2.25% (*T. virens*). All the fatty acids identified in the present work were verified by mass spectrum analysis, which improves the accuracy of identification.

### 3.9. Purification and Characterization of T. brevicompactum Membrane Lipid

Lipids of *T. brevicompactum* were purified by the column chromatography technique. Sequentially, fractions were collected by eluting with different concentrations of chloroform and methanol (9.5:0.5, 9:1, 8:2) in a flow rate of 2 mL/min and were analyzed on TLC plates to identify the presence of different lipid compounds in pure form. Overall, 60 fractions were collected of which fraction 12, 16, 52, and 55 indicates the presence of lipids in pure form with a single spot developed on TLC analysis. Fraction 16 showed a prominent pink-violet-colored orcinol reactive band on a white background at an Rf value of 0.85 in the TLC analysis, indicating the presence of purified sphingolipid [26]. Further, purified fractions were taken to identify the presence of prominent molecules in the sample.

Fatty acid methyl esters (FAMEs) of *T. brevicompactum* lipid confirmed the occurrence of octadecenoic acid. The presence of sugar molecules in the fraction was confirmed by the qualitative Molisch test, which showed a positive result with formation of a purple ring indicating the existence of a sugar fragment. Thin layer chromatography (TLC) with sugar standards, such as glucose, galactose, mannose, arabinose, and xylose, further confirmed the presence of sugar molecule in the purified fraction of *T. brevicompactum* membrane lipid. By visualizing the TLC plates with sugar standards, the sugar molecule present in the purified fraction of *T. brevicompactum* was found to be related to glucose. From the above results, it was clear that the purified sample was a glycospingolipid, which contains a monosaccharide linked with the spingoid (ceremide).

### 3.10. Liquid Chromatography-Mass Spectroscopy (LC-MS) Analysis

Liquid chromatography-mass spectroscopy (LC-MS) provided the structural information of the purified fraction of the *T. brevicompactum* membrane lipid, with a major peak area at retention time of 3.09 min in the positive ionization mode. This peak corresponds to a mass of 342.5 with the predictive molecular formula C_20_H_39_NO_3_. A representative graph of LC-MS with the mass spectrum of the sample is illustrated in Figure 10a,b. The LC-MS results suggested the characteristic signal of sphingoid base moieties.

### 3.11. Fourier Transform Infrared Spectroscopy (FTIR) Analysis

Fourier transform infrared spectroscopy (FTIR) analysis of purified fraction of *T. brevicompactum* membrane lipid confirmed the presence of a functional group of the compound present in the sample (Figure 11). It disclosed a broad peak at 3350 cm^−1^, which is the trait of the O-H stretching form and the N-H stretch in the amine group. The peaks at 2929^−1^ correspond to the asymmetric stretching vibration of -CH_3_, and the asymmetric and symmetric stretching vibrations of -CH_2_. In the spectrum, at 1727 and 1466 cm^−1^, the incidence of the interaction between C=O and C-N groups was specified. Infrared (IR) spectra also showed absorption peaks at 1272, 1121, and 1074 cm^−1^ for a cis double bond, respectively.

### 3.12. Nuclear Magnetic Resonance (NMR) Analysis of the Purified Fraction of T. brevicompactum Membrane Lipid

Results of the NMR analysis of the purified fraction of the *T. brevicompactum* are summarized in Table 6, Table 7 and Table 8. The HH-correlation spectroscopy enabled the connectivity of the long-chain base protons to be traced between C_1_ and C_17_ (refer to Figure 12 for the numbering of the carbon atoms). ^1^H-NMR spectra clearly show the presence of olefinic protons and long chain carbon. The complete assignment of protons is listed in Table 6.

^13^C NMR spectra show the presence of a carbonyl carbon (167 ppm), olefinic carbons (128–130 ppm), and long chain carbons (10–40 ppm) (Figure 13). The complete assignment of carbons is listed in Table 7.

DEPT-135 shows that the purified sample has 13 secondary carbons (>CH_2_), 2 olefinic carbons, and 1 primary carbon (-CH_3_) (Figure 14; Table 8).

2D (NOESY) NMR was performed to know the spatial interaction of the molecule. However, no significant interaction between the molecules was observed. Thus, this indicates that the predicated sphingoid sample in the purified sample of *T. brevicompactum* lipid has no spatial interaction bonds/overlaying in between the molecule.

All the above results of the purified fraction of the membrane lipid of *T. brevicompactum* obtained from the GC-MS, LC-MS, UV-FTIR, and NMR analysis could putatively predict the compound as (E)-N-(1, 3-dihydroxyoctadec-4-en-2yl) acetamide.

The predicted compound is the backbone of sphingolipid. Even though we identified the glucose sugar attachment with putatively predicated sphingolipid of our sample by sugar analysis, the glucose group was not clearly elucidated in LC-MS and NMR studies. This may be because of the possible cleavage of the sugar moiety in the analysis process. So, with all this supporting information, we conclude the work with the putative predication of the purified *T. brevicompactum* membrane lipid as N-acetylsphingosine/C2-ceramide.

Hence, the structure of the (E)-N-(1, 3-dihydroxyoctadec-4-en-2yl) acetamide is:



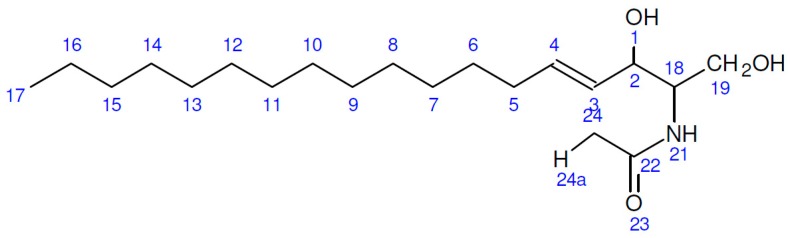



Chemical formula: C_20_H_39_NO_3_Molecular weight: 341.5 g/molCompound IUPAC name: (E)-N-(1, 3-dihydroxyoctadec-4-en-2yl) acetamideCommon name: N-acetylsphingosine/C2-ceramide

### 3.13. Modulation in Defense Enzyme Activities after Priming Pearl Millet Seeds with T. brevicompactum Lipid Nanoemulsion

The temporal modulation of upregulation and downregulation of the enzyme activities in the treated and untreated seedlings with or without pathogen inoculation was examined. Varying patterns of modulation in enzyme activity were observed in seedlings harvested at different time intervals. The sterile distilled water control treatment showed the least enzyme activities at all time intervals tested, confirming the susceptibility of the chosen cultivar to DM infection.

The time course study of LOX in PM seedlings indicated that the initial activity did not differ between the different samples at 0 h.a.i. However, with the increase in the time interval, variation in enzyme activity was observed in treated inoculated seedlings, in which the maximum activity was recorded at 48 h.a.i with 29.5 U. Whereas in the case of control uninoculated seedlings, it showed only marginal changes at all time intervals. In the case of the *T. brevicompactum* lipid nanoemulsion treatments with pathogen inoculation, significant (*p ≤* 0.05) activity with 48 h.a.i was observed, with a three-fold increase over the control. It was observed that after 72 h.a.i, there was no significant change in enzyme activity in the treated seedlings and this was maintained until 96 h.a.i (Figure 15A). However, in the control inoculated seedlings, the highest activity was noticed at 24 h.a.i with 6.5 U and it was not significantly (*p ≤* 0.05) different from the control uninoculated seedlings.

Allene oxide synthase (AOS) activities were examined temporally in the *T. brevicompactum* lipid nanoemulsion-treated seedlings with pathogen-inoculated and uninoculated seedlings. The spectrophotometric assay revealed that 13-HPOT was metabolized earlier in the treated inoculated seedlings when compared with the uninoculated seedlings of the same treatment. The highest significant (*p ≤* 0.05) activity was observed in treated seedlings at 24 and 48 h.a.i compared to other treatments. Activity in the treated seedlings was maintained in a moderate parameter till 72 h.a.i, by which a 1.5-fold increase over control inoculated seedlings was observed. After 72 h.a.i, a decline in AOS activities was noticed in all the treated seedlings (Figure 15B).

### 3.14. Gene Expression Studies by Transcriptional Analysis

The relative expression pattern of LOX, AOC, and α-DOX in 7042S control PM cultivar in response to *S. graminicola* was analyzed by qRT-PCR. A significantly (*p ≤* 0.05) higher transcript accumulation at the basal level was observed in the *T. brevicompatum* membrane lipid nanoemulsion treatment compared to the susceptible control. There is no (*p ≤* 0.05) significant relative change seen in the control samples upon inoculation with *S. graminicola*. The expression profile revealed an increase in mRNA accumulation in the *T. brevicompatum* membrane lipid nanoemulsion treatment, which was nearly a 4- to 5-fold increase over the control (Appendix A and Figure 16).

A remarkable difference in the expression levels of different genes in PM in response to the DM pathogen indicates a role of these genes in PM–DM interaction. The expression pattern obtained from the control inoculated treatment was relatively low-level expression compared to others. With this regard, the *T. brevicompatum* membrane lipid nanoemulsion treatment was significantly (*p ≤* 0.05) effective in upregulating the defense gene expression, which means a significant (*p ≤* 0.05) increase in expression was observed in the susceptible 7042S cultivar at different time points upon treatment with the membrane lipid nanoemulsion.

The expression profile revealed an increase in mRNA accumulation in the *T. brevicompatum* membrane lipid nanoemulsion treatment, in which a 3.8-fold increase over the control in was observed in *LOX* gene expression, 4-fold increase over the control in *AOC* gene expression, and 4.1-fold increase over the control was observed in *α-DOX* gene expression analysis. These significant (*p*
*≤* 0.05) changes support the effective role of membrane lipid nanoemulsion in enhancing the defense gene response in the PM cultivar, thereby inducing disease resistance.

### 3.15. GC-MS Analysis of JA and MeJA

In the GC-MS analysis, JA was separated with a retention time of 10.52 min in the negative mode of the ionization potential (*m*/*z* of 224) and MeJA was separated with a retention time of 21.54 min in the positive mode of the ionization potential (m/z of 225). Therefore, the GC peaks of trimethylsilylated JA and MeJA were well separated from those of the trimethylsilylated internal standards, allowing accurate quantification of JA and MeJA.

In *T. brevicompactum* lipid nanoemulsion-treated seedlings, different concentrations of JA and MeJA were observed in nmol concentration. It showed a significantly (*p ≤* 0.05) higher concentration of JA (11.57 nmol/g) and MeJA (15.39 nmol/g) compared with the control treatments. However, the JA and MeJA levels were found to be below the detection limits in the control treatments (Figure 17).

## 4. Discussion

Oils are not soluble in water; this character of oil is a problem in application aspects. Seed priming is a process, which occurs through absorption of the elicitor molecule. So, the compounds used for the seed priming process must be solubilized and dispersed in water. Oils are volatile complex mixtures with a wide range of biological activities, including repellent, insecticidal, and larvicidal properties [38]. With this context, an O/W nanoemulsion of extracted crude lipid from *Trichoderma* spp. will have to overcome solubility in water. Factors like surfactant and emulsification procedures will significantly influence the formation of emulsion. Further, synthesized lipid emulsion of *Trichoderma* spp. was analyzed for its physico-chemical characteristic properties. Lipid emulsion obtained 1 h after sonication was subjected to particle size analysis by the zeta potential and dynamic light scattering process (DLS). The particle size in the emulsified sample was found in the range 5–51 nm, with a polydispersity index of 0.156 to 0.333 (Table 1). The formation of emulsion and its surface methodology signifies the influence of the lipid, emulsifier concentration, emulsification time, and formation of the droplet size. Generally, the smaller the droplet size, the greater the stability of the emulsification [39].

An ultrasonic method is usually preferred compared to others because of its low-cost equipment, smaller footprint, easy cleaning, and good service [20]. Excessive surfactant may result in a lower diffusion rate of the surfactant, and a lower content might lead to lower diffusion and amalgamation of the emulsion droplets [40]. The lipid content and ultrasonication time influences the droplet size at a fixed surfactant content of 5%. The stability of the emulsion is influenced by the ultrasonication time interval, as it affects the droplet size distribution and surfactant adsorption rate on the droplet surface [40].

In emulsion, sedimentation is a reversible destabilization phenomenon, whereas variation of the droplet size is an irreversible process [41]. The polydispersity index (PDI) was found in the range of 0.156 to 0.333, which means the distribution of the droplet size is in the narrow range. Further, the zeta potential was found in the range of +17.8 to −28.2 mv and pH of 7. Optimal nanoemulsion with low turbidity (600 nm absorbance = 0.36 ± 0.01) signifies a smaller droplet size [42]. This fact makes nanoemulsion suitable for its incorporation into different systems without altering the visual quality. Furthermore, there was no significant difference noticed of the polydispersity during the incubation period. Micelles are continuously disintegrating and reassembling, being in dynamic equilibrium with individual surfactant molecules. The optimum condition facilitated the formation of the stable O/W emulsion with a minimum droplet size, i.e., within the nanometer range. In wheat bran emulsion, it was observed that thee surfactant to oil ratio is relatively high [17], but practically, due to the economic and rigid basis, it was not a desirable application.

The interaction between the pathogen and host induces challenges in cell metabolism, primarily in enzyme activities, including lipoxygenase [35]. In the membrane lipid nanoemulsion treatment, significant (*p ≤* 0.05) disease protection was achieved from *T. brevicompactum* against the DM pathogen. Further, time interval studies using the best treatment were carried out to know the induction of resistance. It was also observed that *T. brevicompactum*-primed plants are capable of triggering defense responsive genes in a susceptible cultivar in a similar way to that of the resistant cultivar treated with *T. brevicompactum*; however, the level of gene expression was low in susceptible cultivars. In this context, our results suggest that the nanoemulsion containing 5% (*w*/*w*) of crude membrane lipid of *Trichoderma* spp., 5% (*w*/*w*) of polysorbate 80 (*w*/*w*) and 90% (*w*/*w*) of water can be considered a promising biocontrol formulation for the control of oomycetes pathogen *S. graminicola* of PM. Glycosphingolopids isolated from various species of *F. oxysporum* also induced resistance against various *Fusarium* diseases. Extraction and purification of cerebrosides from different fungi have been reported earlier [43,44]. Previously, Naveen et al. [26] studied the cerebroside-mediated disease resistance in chili by eliciting the production of defense-related enzymes, accretion of H_2_O_2_, and accumulation of capsidiol.

The results of the present study suggest that the droplet size of the membrane lipid nanoemulsion of *Trichoderma* spp. is inversely proportional to the induction of disease resistance in PM. Further, membrane lipid nanoemulsion from *T. brevicompactum* is effective in reducing DM disease of PM. Therefore, this study points to a potential integrated approach for the future management of DM disease in PM. The results of the present study illustrate potential efficacy data of the lipid nanoemulsion of *Trichoderma* spp. to manage the DM pathogen. Additionally, it opens a new avenue, where *Trichoderma* membrane lipid nanoemulsion formulations can be successfully employed for plant disease management.

Sphingolipids have a common structural characteristic, having backbones called “long-chain-” or “sphingoid” bases, represented by sphingosine, (2S,3R,4E)-2-aminooctadec-4-ene-1,3-diol (also called (E)-sphing-4-enine), and N-Acyl-sphingoid bases (ceramides). Acylation of the amino group of sphingoid bases with a fatty acid produces compounds generally referred as ceramides-2, specifically termed as N-acylsphingosines or dihydroceramides for N-acylsphinganines and 4-hydroxyceramides or phytoceramides for N-acyl-4-hydroxysphinganines [45,46].

In the present work, FAME analysis of six different *Trichoderma* spp. indicated the presence of nine fatty acids in varying quantity, in which oleic acid was found in the maximum quantity. Further, sphingolipid from the *T. brevicompactum* lipid was extracted and it was characterized by chemical, spectrometric, and spectroscopic analysis. The lipid composition of six different *Trichoderma* spp. was identified by FAME analysis. In all the *Trichoderma* spp., oleic acid was found in the highest quantity except in *T. harzianum*, in which palmitoleic acid was recorded in maximum quantity. FAME analysis provided the predictive fatty acid long chain of the *T. brevicompactum* membrane lipid. Furthermore, LC-MS analysis helps to know the molecular mass of the purified sample of *T. brevicompactum* membrane lipid with a putative predicated molecular formula. NMR analysis provided evidence of ^1^H protons and ^16^C chains in the purified sample. DEPT-135 discloses the occurrence of 13 secondary carbons, 2 olefinic carbons, and 1 primary carbon in the sample. With all the above supporting information, the compound is putatively predicated in the purified fraction of *T. brevicompactum* as N-acetylsphingosine/C2-ceramide. The presence of the C2-ceramide backbone structure has been reported previously in some fungal species [47].

In the present study, fatty acid profiling of six different *Trichoderma* spp. illustrates the presence of nine different types of fatty acid, in which oleic acid was found in the maximum quantity and its presence was observed both in the *-cis* and *-trans* form. Further, the membrane lipid of *T. brevicompactum* was purified and characterized to know the occurrence of prominent molecules in *T. brevicompactum,* in which N-acetylsphingosine was identified, which mimics the microbial-derived molecular patterns in inducing disease resistance in PM against DM disease. Hence, an increased level of resistance in pear millet seedlings, primed with *T. brevicompactum* lipid nanoemulsion, was observed against DM disease.

PM–DM pathogen interaction studies are significantly important to characterize the structural elements required for recognition, and receptor molecules to understand the molecular basis for the perception and transduction of elicitor signals. In the present study, we focused on the modulation of oxylipin pathways’ enzymatic reaction in response to DM pathogen infection. Further, time interval studies using the best treatment of elicitors, i.e., lipid nanoemulsion of *T. brevicompactum*, were carried out to study the induction of resistance. In the LOX assay, significant activity (*p ≤* 0.05) was noticed at 48 h.a.i. and in the AOS assay, the highest activity was recorded at 24 h.a.i, confirming the induction and durability of resistance. The ISR is associated with increased activity of defense-related enzymes, such as PAL, POX, and β-1, 3-glucanase and LOX [9,12,35]. Hindumathy et al. [48] used glucan elicitors derived from the cell wall of yeast to elicit disease resistance. Here, glucan elicitor seed treatment significantly enhanced defense-related enzymes, such as tyrosine ammonia lyase, and poly phenol oxidase, upon challenge inoculation. In our studies, higher activity of LOX and AOS was observed in seedlings treated with elicitor and also the activity of these defense molecules was significantly (*p ≤* 0.05) higher in seedlings after challenge inoculation. The above findings suggest the role for these defense-related molecules in the expression of induced resistance by lipid nanoemulsion elicitor against *S. graminicola* invasion in PM.

Elevated levels of defense-related enzymes and signaling molecules recorded in treated PM seedlings and their further increase after *S. graminicola* infection indicates that seed treatment of PM with nanoemulsion of *T. brevicompactum* lipid creates an incompatible environment for infection, proliferation, and sporulation by *S. graminicola*, which leads to disease suppression [8,35,36]. The gene expression profile demonstrates a boosting up in mRNA accumulation in the host after *T. brevicompatum* membrane lipid nanoemulsion treatment, which was nearly a 4- to 5-fold increase over the control. A noteworthy discrepancy was observed in the expression levels of different genes in PM in response to the DM pathogen, signifying the function of these genes in PM–DM pathogen interactions. Torre-Hernandez et al. [49] illustrated plant defense triggered by sphingolipid molecules, i.e., fumonisin B1 stimulates activation of nuclease and salicylic acid accumulation through sphingoid long-chain base build-up in germinating maize.

Jasmonic acid (JA) as a regulator has a significant role in the growth of higher plants and activates defense response genes under stress conditions [9]. The result shows that LOX activity increases 3-fold to a maximal level approximately 48 h.a.i. Activity of the AOS enzyme was significant at 24 h.a.i, with a 1.5-fold increase over the control, and a decline in enzyme activity was observed after 72 h.a.i. In both the enzyme assays, prominent activity was observed at a time interval of 48 h.a.i. On this basis, a 48 h.a.i time interval of treatment with or without pathogen inoculation was selected for estimation and quantification of JA and MeJA by GC-MS analysis to study the upregulation and downregulation of these compounds in treated seedlings. Jasmonic acid (JA) is a cyclic derivative of linolenic acid and its methyl ester, MeJA, has been considered a key player in inducing specific defense responses and appear to also be concerned in a few pathogen-induced defense reactions [9]. The result of our studies suggests an increase in JA and MeJA in treated seedlings with significant (*p ≤* 0.05) accumulation of signaling molecules compared to the control inoculated treatment. Accumulation of JA and MeJA in PM seedlings was well correlated with the enhanced enzyme activity and decrease in DM disease incidence.

The above findings suggest the role for these defense-related molecules in the expression of induced resistance by lipid nanoemulsion elicitor treatment against *S. graminicola* invasion in PM. During ISR, the role of enzymes like chitinases, peroxidase, PAL, LOX, and β-1, 3-glucanase is well studied [9,12,35,36]. In this context, our results suggest that the lipid nanoemulsion of *T. brevicompactum* can be considered a promising biocontrol formulation for the control of the oomycetes pathogen, *S. graminicola*, of pearl millet.

## Figures and Tables

**Figure 1 biomolecules-10-00025-f001:**
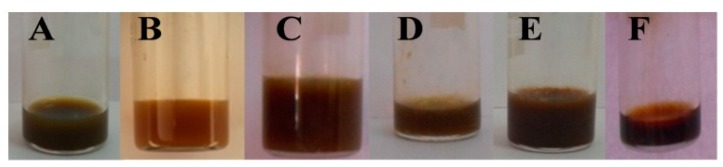
Extraction of lipids from *Trichoderma spp.*
**A**—*T. asperellum*; **B**—*T. harzianum*; **C**—*T. virens*; **D**—*T. longibrachiatum*; **E**—*T. atroviride*; **F**—*T. brevicompactum*.

**Figure 2 biomolecules-10-00025-f002:**
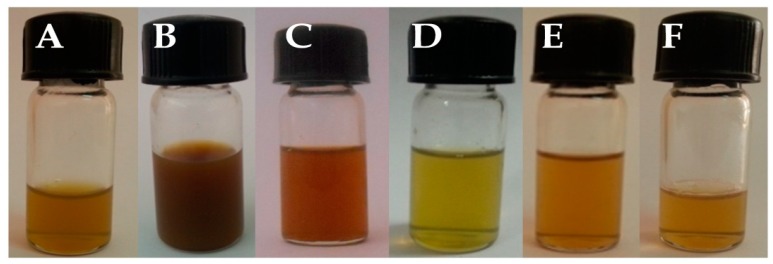
Lipid nanoemulsions of *Trichoderma spp.*
**A**—*T. asperellum*; **B**—*T. harzianum*; **C**—*T. virens*; **D**—*T. longibrachiatum*; **E**—T. atroviride; **F**—*T. brevicompactum*.

**Figure 3 biomolecules-10-00025-f003:**
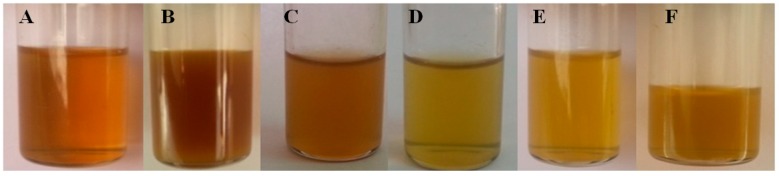
Lipid nanoemulsions of *Trichoderma* spp. after 60 days of incubation at 4 °C. **A**—*T. asperellum*; **B**—*T. harzianum*; **C**—*T. virens*; **D**—*T. longibrachiatum*; **E**—*T. atroviride*; **F**—*T. brevicompactum*.

**Figure 4 biomolecules-10-00025-f004:**
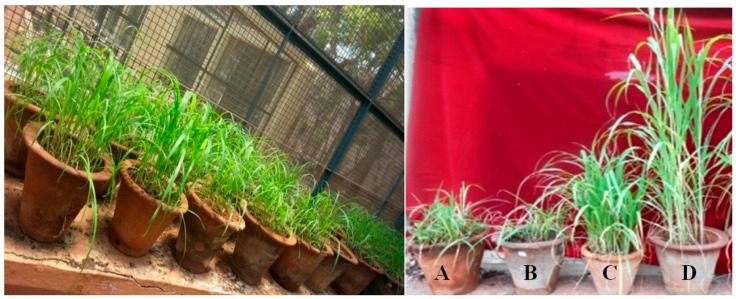
Greenhouse studies of extracted lipids from *Trichoderma* spp. inducing downy mildew disease resistance in pearl millet. **A**—Sterile distilled water control; **B**—Tween 80; **C**—Metalaxyl (6 g/kg seeds); **D**—*T. brevicompactum* lipid nanoemulsion treatment.

**Figure 5 biomolecules-10-00025-f005:**
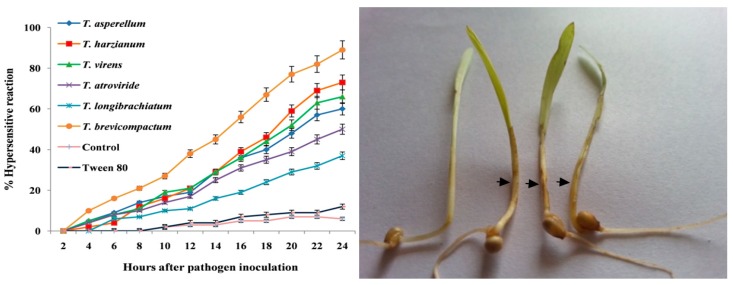
Hypersensitive reaction observed in the nanoemulsion of *Trichoderma* membrane lipid-treated pearl millet seedlings. Arrows indicate the accumulation of hypersensitive reaction in pearl millet seedlings.

**Figure 6 biomolecules-10-00025-f006:**
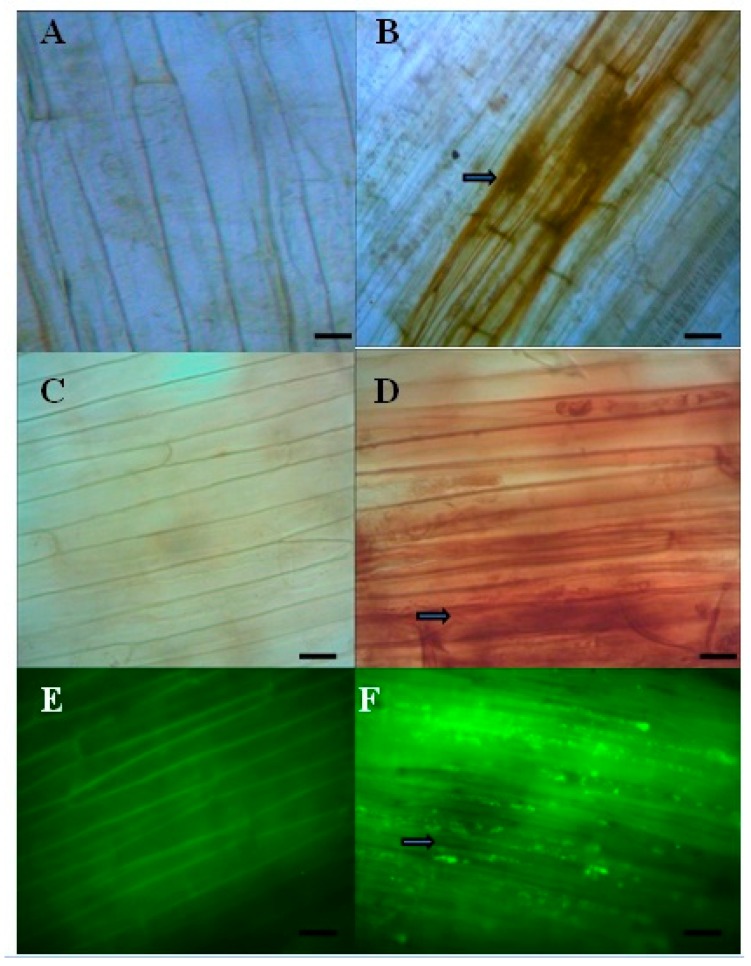
Histological studies of *T. brevicompactum* nanoemulsion-treated seedlings. Lignin deposition, **A**—control; **B**—treated; H_2_O_2_ deposition; **C**—control; **D**—treated seedlings; Callose deposition; **E**—control; **F**—treated seedlings.

**Figure 7 biomolecules-10-00025-f007:**
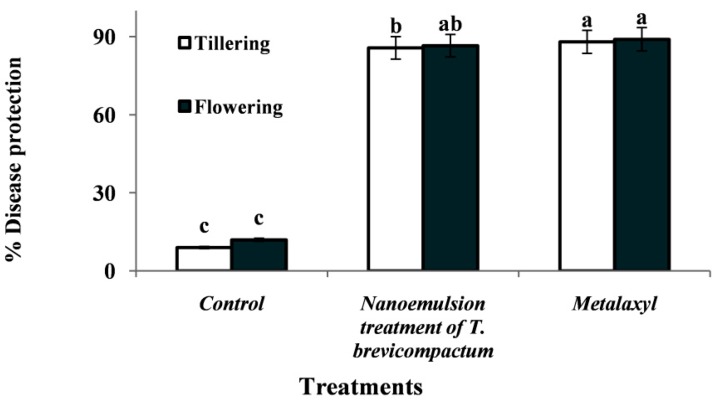
Durability of induced resistance in pearl millet against downy mildew disease by *T. brevicompactum* lipid nanoemulsion treatment.

**Figure 8 biomolecules-10-00025-f008:**
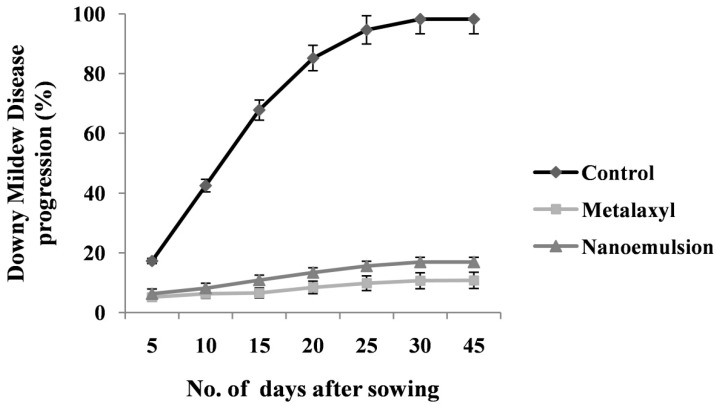
Effect of induced resistance by *T. brevicompactum* lipid nanoemulsion treatment against downy mildew disease of pearl millet under field conditions.

**Figure 9 biomolecules-10-00025-f009:**
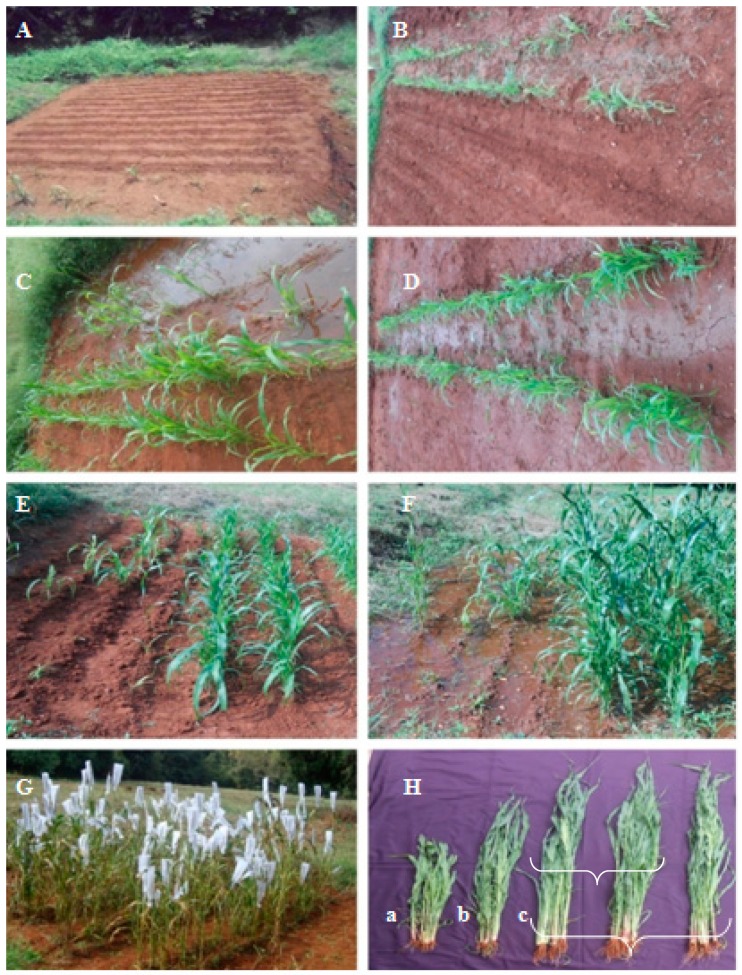
Field studies of *T. brevicompactum* lipid nanoemulsion treatment induced resistance against downy mildew disease of pearl millet; **A**—Downy mildew sick plot; **B**—Infector rows, 21 days after sowing; **C**–**G**—*T. brevicompactum* lipid nanoemulsion treatment; **D**—30 days after sowing; **E**—40 days after sowing; **F**—55 days after sowing; **G**—60 days after sowing; **H**—Seedling growth parameters under field conditions, **a**—Control; **b**—Metalaxyl; **c**—*T. brevicompactum* lipid nanoemulsion treatment. *Trichoderma* spp. (between the columns).

**Figure 10 biomolecules-10-00025-f010:**
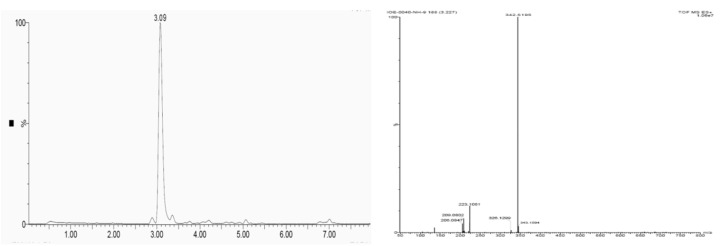
(**a**) LC-MS chromatogram and (**b**) mass spectrum of the purified *T. brevicompactum* lipid sample.

**Figure 11 biomolecules-10-00025-f011:**
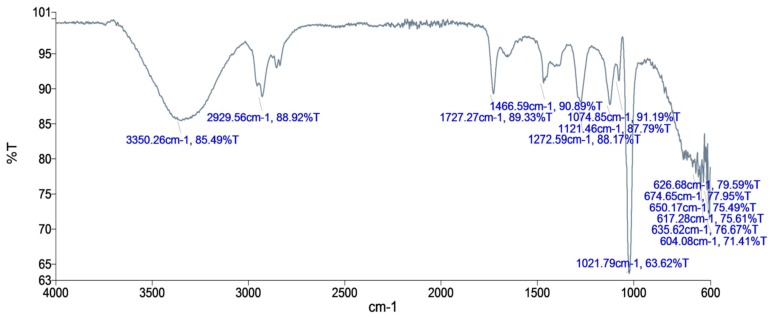
FTIR spectrum of the purified *T. brevicompactum* lipid sample.

**Figure 12 biomolecules-10-00025-f012:**
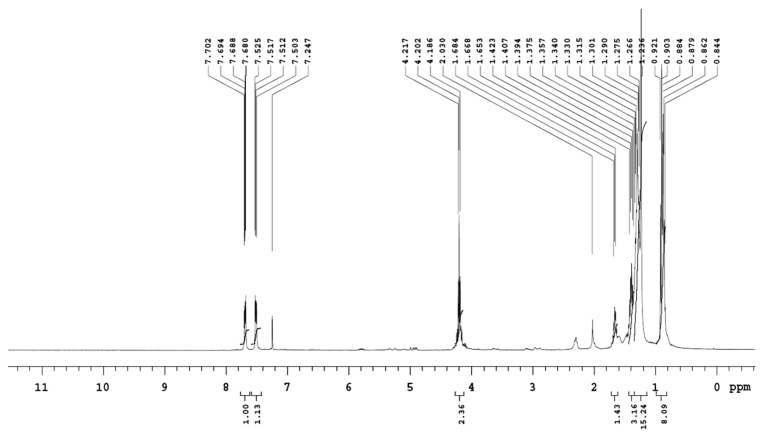
^1^H-NMR spectrum of the purified fraction of *T. brevicompactum* lipid.

**Figure 13 biomolecules-10-00025-f013:**
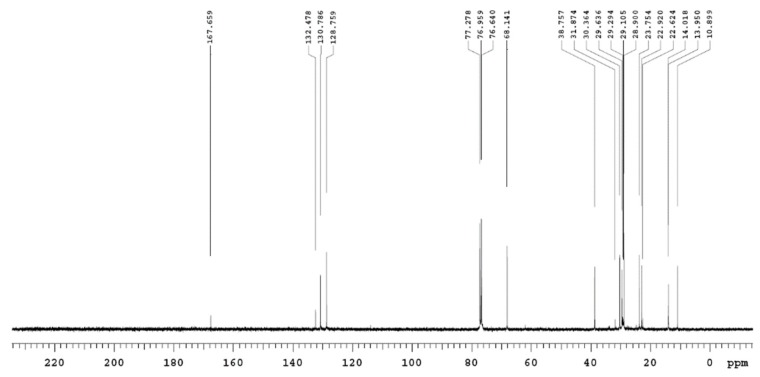
^13^Carbon -NMR spectrum of the purified fraction of *T. brevicompactum* lipid.

**Figure 14 biomolecules-10-00025-f014:**
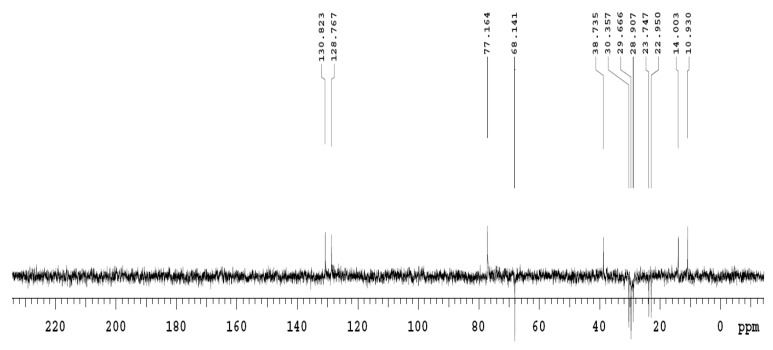
DEPT-135 NMR spectrum of the purified fraction of *T. brevicompactum* lipid.

**Figure 15 biomolecules-10-00025-f015:**
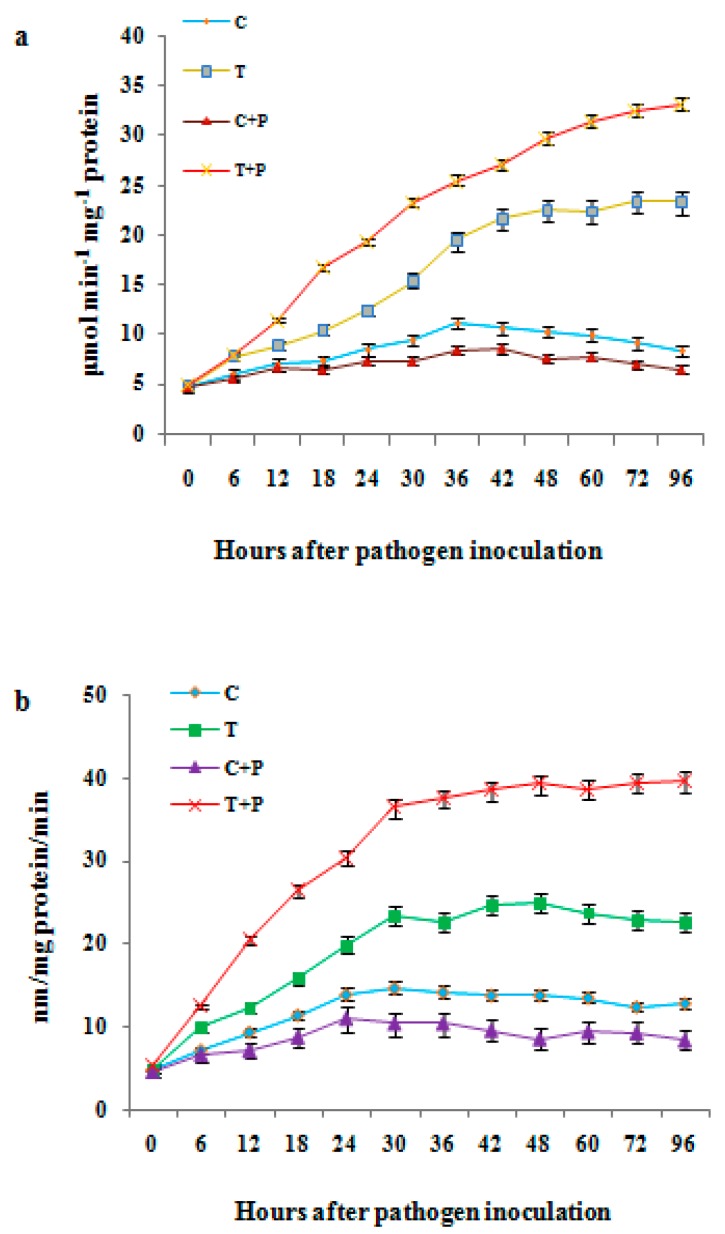
Temporal pattern of the accumulation of defense-related enzymes in pearl millet seedlings upon seed treatment with *T. brevicompactum* lipid nanoemulsion. (**A**) Lipoxygenase LOX (**B**) Allene oxide synthase AOS. Data of enzyme activity are means ± SE of three different experiments. The values were the means of three replicates of three different experiments. C—SDW control; C + P—SDW control + pathogen; T—*T. brevicompactum* lipid nanoemulsion; T + P—*T. brevicompactum* lipid nanoemulsion + pathogen.

**Figure 16 biomolecules-10-00025-f016:**
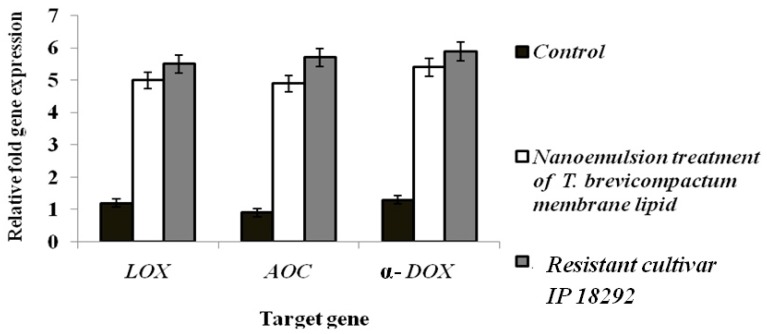
Relative expression levels of gene transcripts in two-day-old pearl millet seedlings in response to pathogen inoculation. Specific gene expression levels were measured by qPCR and normalized to the constitutive reference gene. The values represented are means of a single experiment performed in triplicates. The bars indicate ± SE.

**Figure 17 biomolecules-10-00025-f017:**
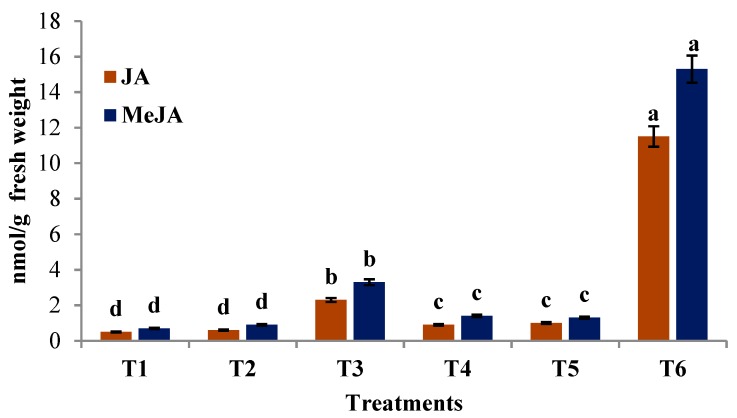
Systemic production of jasmonic acid and methyl jasmonate in pearl millet seedlings upon seed priming with *T. brevicompactum* lipid nanoemulsion. T1—control; T2—5% Tween 80; T3—*T. brevicompactum* lipid nanoemulsion; T4—control pathogen; T5—5% Tween 80 + pathogen; T6—*T. brevicompactum* lipid nanoemulsion + pathogen.

**Table 1 biomolecules-10-00025-t001:** Physico-chemical characterization of membrane lipid nanoemulsion of *Trichoderma* spp.

Parameters	*Ta*	*Th*	*Tv*	*Tl*	*Tr*	*Tb*
**Droplet size (nm)**	8.81	50	7.91	5.94	51	6.11
**Shape**	Spherical	Spherical	Spherical	Spherical	Spherical	Spherical
**pH**	7	7	7	7	7	7
**Viscosity (cp)**	0.810	0.811	0.808	0.811	0.812	0.806
**Zeta Potential**	−7.4 mv	−4.6 mv	+17.8 mv	7.3 mv	−28.2 mv	−7.5 mv
**Polarity**	Negative	Negative	Positive	Positive	Negative	Negative
**Absorbance (600 nm)**	0.014	0.006	0.028	0.053	0.017	0.067
**Polydispersity index**	0.333	0.260	0.161	0.206	1.998	0.156

*Ta*—*T. asperellum*; *Th*—*T. harzianum*; *Tv*—*T. virens*; *Tl*—*T. longibrachiatum*; *Tr*—*T. atroviride*; *Tb*—*T. brevicompactum*.

**Table 2 biomolecules-10-00025-t002:** Effect of seed priming with extracted lipids from *Trichoderma* spp. on seed germination and seedling vigor of pearl millet.

Treatment		*T. asperellum*		*T. harzianum*		*T. virens*		*T. atroviride*		*T. longibrachiatum*		*T. brevicompactum*	
	Conc. µg/mL	% G	SV	%G	SV	% G	SV	% G	SV	% G	SV	% G	SV
	**I Method**												
**I Method**	50	89 ± 0.67 ^ab^	1596 ± 16.70 ^ab^	89 ± 0.67 ^ab^	1566 ± 13.75 ^b^	90 ± 0.58 ^ab^	1616 ± 60.18 ^a^	90 ± 0.88 ^abc^	1608 ± 10.49 ^b^	89 ± 0.88 ^a^	1540 ± 13.53 ^a^	90 ± 0.58 ^abc^	1587 ± 28.34 ^ab^
	100	91 ± 1.00 ^ab^	1610 ± 22.91 ^ab^	92 ± 0.88 ^ab^	1619 ± 51.29 ^b^	90 ± 1.00 ^ab^	1632 ± 39.30 ^a^	91 ± 0.58 ^ab^	1671 ± 37.59 ^ab^	89 ± 0.58 ^a^	1558 ± 58.36 ^a^	92 ± 0.58 ^ab^	1623 ± 64.95 ^ab^
	150	90 ± 1.45 ^ab^	1615 ± 63.55 ^ab^	91 ± 0.67 ^ab^	1601 ± 61.23 ^b^	89 ± 1.15 ^ab^	1603 ± 46.28 ^a^	91 ± 0.58 ^ab^	1661 ± 34.79 ^b^	91 ± 0.67 ^a^	1614 ± 19.67 ^a^	90 ± 1.45 ^abc^	1614 ± 71.92 ^ab^
**II Method**		**II Method**											
**Lipid Nanoemulsion**		92 ± 0.88 ^a^	1815 ± 61.20 ^a^	93 ± 0.33 ^a^	1879 ± 21.74 ^a^	92 ± 0.88 ^a^	1742 ± 53.76 ^a^	92 ± 0.33 ^a^	1795 ± 51.76 ^a^	91 ± 0.33 ^a^	1747 ± 73.00 ^a^	93 ± 0.58 ^a^	1921 ± 112.64 ^a^
**SDW Control**		88 ± 0.58 ^b^	1525 ± 80.16 ^b^	88 ± 0.58 ^c^	1525 ± 80.16 ^b^	88 ± 0.58 ^b^	1525 ± 80.16 ^a^	88 ± 0.58 ^c^	1525 ± 80.16 ^b^	88 ± 0.58 ^a^	1525 ± 80.16 ^a^	88 ± 0.58 ^c^	1525 ± 80.16 ^b^
**Buffer**		89 ± 1.20 ^ab^	1552 ± 86.59 ^b^	89 ± 1.20 ^ab^	1552 ± 86.59 ^b^	89 ± 1.20 ^ab^	1552 ± 86.59 ^a^	89 ± 1.20 ^bc^	1552 ± 86.59 ^b^	89 ± 1.20 ^a^	1552 ± 86.59 ^a^	89 ± 1.20 ^bc^	1552 ± 86.59 ^ab^
**Tween 80**		89 ± 1.15 ^ab^	1535 ± 97.78 ^b^	89 ± 1.15 ^ab^	1535 ± 97.78 ^b^	89 ± 1.1 ^ab^	1535 ± 97.78 ^a^	89 ± 1.15 ^bc^	1535 ± 97.78 ^b^	89 ± 1.15 ^a^	1535 ± 97.78 ^a^	89 ± 1.15 ^bc^	1535 ± 97.78 ^b^
**Metalaxyl ***		89 ± 0.88 ^ab^	1547 ± 53.42 ^b^	89 ± 0.88 ^c^	1547 ± 53.42 ^b^	89 ± 0.88 ^ab^	1547 ± 53.42 ^a^	89 ± 0.88 ^bc^	1547 ± 53.42 ^b^	89 ± 0.88 ^a^	1547 ± 53.42 ^a^	89 ± 0.88 ^c^	1547 ± 53.42 ^ab^

Values are the mean within the column sharing the same letters are not significantly different according to Tukey’s HSD at *p ≤* 0.05. * Metalaxyl was used as seed dressing at of 6 g/kg seeds. % G—Percent germination; SV—Seedling vigor; SDW—Sterile distilled water; Buffer-Potassium phosphate buffer (20 mM) pH 6.5, containing 0.1% Tween 20.

**Table 3 biomolecules-10-00025-t003:** Effect of *Trichoderma* membrane lipid-induced downy mildew disease resistance in pearl millet under greenhouse conditions.

	% Disease Protection						
Treatment	Conc. µg/mL	*T.a*	*T.h*	*T.v*	*T.l*	*T.r*	*T.b*
**I Method**	**I Method**						
	50	39.2 ± 0.38 ^j,k^	33.9 ± 0.64 ^k^	54.0 ± 1.39 ^e,f,g^	34.2 ± 1.39 ^k^	43.9 ± 1.56 ^h,i,j^	52.5 ± 0.84 ^e,f,g^
	100	50.7 ± 1.47 ^f,g,h^	39.8 ± 1.44 ^j,k^	63.9 ± 2.65 ^b,c,d^	41.3 ± 1.77 ^i,j,k^	53.3 ± 1.93 ^e,f,g^	68.6 ± 2.07 ^b,c^
	150	44.0 ± 2.68 ^h,i,j^	35.1 ± 1.89 ^k^	49.3 ± 1.89 ^ghi^	37.5 ± 1.10 ^j,k^	37.3 ± 0.98 ^j,k^	60.2 ± 2.06 ^c,d,e^
	**II Method**						
**II Method Lipid Nanoemulsion**		58.2 ± 2.08 ^d,e,f^	45.8 ± 1.33 ^g,h,I,j^	70.3 ± 0.76 ^b^	37.8 ± 1.06 ^j,k^	64.4 ± 1.82 ^b,c,d^	84.8 ± 0.68 ^a^
**SDW Control**		^--n^					
**Phosphate Buffer**		11.4 ± 0.59 ^l,m^					
**Tween 80 (5%)**		14.5 ± 1.84 ^l^					
**Metalaxyl ***		89.5 ± 0.78 ^a^					

Values are the mean within the column sharing the same letters are not significantly different according to Tukey’s HSD at *p ≤* 0.05. * Metalaxyl was used as seed dressing at of 6 g/kg seeds. SDW—sterile distilled water; Potassium phosphate buffer (20 mM) pH 6.5, containing 0.1% Tween 20. *T.a*—*T. asperellum*; *T.h*—*T. harzianum*; *T.v*—*T. virens*; *T.l*—*T. longibrachiatum*; *T.r*—*T. atroviride*; *T.b*—*T. brevicompactum.*

**Table 4 biomolecules-10-00025-t004:** Effect of seed treatment with *T. brevicompactum* lipid nanoemulsion on growth parameters of pearl millet plants under field conditions.

Growth Parameters	Control	Treatment (*T. brevicompactum* Lipid Nanoemulsion)	Metalaxyl *
Plant height (cm)	59.3 ± 0.63 ^c^	76.2 ± 0.37 ^a^	70.5 ± 0.33 ^b^
No. of productive ear head	3.1 ± 0.25 ^b^	3.8 ± 0.24 ^a^	3.3 ± 0.15 ^a,b^
Length of ear head (cm)	10.6 ± 0.16 ^c^	13.5 ± 0.15 ^a^	11.4 ± 0.17 ^a,b^
Girth of ear head (cm)	3.3 ± 0.11 ^b^	4.3 ± 0.16 ^a^	3.7 ± 0.16 ^b^
**1000-seedweight (grams)**	8.5 ± 0.09 ^b^	10.9 ± 0.07 ^a^	10.2 ± 0.11 ^a^

Values are means of three independent replicates; Means followed by the same letter(s) within the column are not significantly different according to Tukey’s HSD. * Metalaxyl was used as seed dressing at of 6 g/kg seeds.

**Table 5 biomolecules-10-00025-t005:** Composition of fatty acids in different *Trichoderma* spp.

Peak No.	Fatty Acid	Mass	RT	*T.a*	*T.h*	*T.v*	*T.l*	*T.r*	*T.b*
1	Lauroleic Acid (C12:1)	198.2	4.269	2.31 ^b^	2.98 ^b^	0.62 ^c^	2.69 ^b^	0.67 ^c^	6.68 ^a^
2	Myristic acid (C14:0)	228.4	6.230	5.77 ^a^	6.88 ^a^	1.12 ^c^	2.76 ^b^	0.66 ^c^	6.09 ^a^
3	Pentadeconic acid (C15:0)	242.4	7.562	2.00 ^a^	0.66 ^b^	1.18 ^a^	1.97 ^a^	0.53 ^b^	1.44 ^a^
4	Palmitic acid (C16:0)	256.4	8.346	0.53 ^d^	3.81 ^a^	1.60 ^c^	2.68 ^b^	0.97 ^d^	1.25 ^c^
5	Palmitoleic acid (C16:1)	254.4	8.921	6.91 ^b^	35.07 ^a^	0.75 ^c^	8.91 ^b^	6.18 ^b^	6.68 ^b^
6	Stearic acid (C18:0)	284.5	11.707	27.15 ^a^	26.07 ^b^	28.45 ^a^	28.28 ^a^	28.3 ^a^	19.72 ^c^
7	Linoleic acid (ω6, C18:2)	280.4	13.838	1.88 ^b^	7.24 ^a^	0.66 ^c^	1.25 ^b^	0.73 ^c^	0.67 ^c^
8	Oleic acid (C18:1, cis-9)	282.5	16.154	26.43 ^b^	5.45 ^d^	28.83 ^a^	15.96 ^c^	29.02 ^a^	24.15 ^b^
9	Oleic acid (C18:1, trans-13)	282.5	16.429	23.24 ^b^	8.03 ^c^	25.48 ^b^	35.06 ^a^	32.34 ^a^	30.37 ^a^
10	Arachidonic acid (C20:2)	304.4	17.284	3.78 ^c,d^	3.81 ^e,f^	11.31 ^c^	0.43 ^g,h^	0.52 ^d,e^	2.95 ^h^
	Total saturated fatty acids (SFA)			35.45 ^a^	37.42 ^a^	32.35 ^b^	35.69 ^a^	30.54 ^b^	28.5 ^c^
	Total monounsaturated fatty acids (MUFA)			58.89 ^b^	51.53 ^c^	55.68 ^b^	62.62 ^a^	68.21 ^a^	67.88 ^a^
	Total polyunsaturated fatty acids (PUFA)			5.66 ^b^	11.05 ^a^	11.97 ^a^	1.68 ^c^	1.25 ^c^	3.62 ^b^
	PUFA: SFA			0.16 ^b^	0.29 ^a^	0.37 ^a^	0.05 ^c^	0.041 ^c^	0.13 ^b^
	PUFA: MUFA			0.11 ^b^	0.21 ^a^	0.22 ^a^	0.027 ^c^	0.02 ^c^	0.05 ^c^
	Unsaturated fatty acid (UFA)			64.55 ^c^	62.58 ^c^	67.65^b^	64.3 ^c^	69.46 ^a^	71.5 ^a^
	SFA:UFA			0.55 ^a^	0.60 ^a^	0.48 ^b^	0.56 ^a^	0.44 ^b^	0.40 ^b^
	Total lipids			3.23 ^a^	3.18 ^a^	2.25 ^b^	2.32 ^b^	2.29 ^b^	2.26 ^b^

RT—retention time; *T.a*—*T. asperellum*; *T.h*—*T. harzianum*; *T.v*—*T. virens*; *T.l*—*T. longibrachiatum*; *T.r*—*T. atroviride*; *T.b*—*T. brevicompactum*. Values are percentages of the total fatty acids, from an average of three replicates. Different superscript letter indicates statistically significant (*p ≤* 0.05) differences among different.

**Table 6 biomolecules-10-00025-t006:** Summary of proton (^1^H) NMR chemical shifts (ppm).

δ-^1^H	Multiplicity	No. of Protons	Position of Carbon Chain
**7.7**	m	1H	C_3_
**7.5**	m	1H	C4
**4.2**	t	2H	C19
**1.65**	t	1H	C2
**1.35**	m	1H, 2H	C18,C5
**1.27**	m	15	C8, C9, C13, C14, C15, C16, C24
**1.3**	m	2H	C5
**1.3**	m	4H	C6, C7
**0.87**	m	3H	C17
**0.9**	m	6H	C10,11,12

**Table 7 biomolecules-10-00025-t007:** Summary of ^13^C-NMR chemical shifts (ppm).

δ Value	No. of Carbon Atoms	Expected Carbon	Position of Carbon Chain
**167.659**	1C	C=O	C22
**132.478**	1C	-C-OH	C2
**130.786**	1C	-C=C-	C3
**128.759**	1C	-C=C-	C4
**68.141**	1C	-CH_2_-0H	C19
**38.757**	1C	-CH-	C18
**30.364**	2C	-CH_2_-	C6,C7
**29.636**	2C	-CH_2_-	C8,C9
**28.900**	2C	-CH_2_-	C10,C11
**23.754**	2C	-CH_2_-	C12,C13
**22.920**	2C	-CH_2_-	C14,C15
**13.950**	1C	-CH_2_-	C16
**10.899**	1C	-CH3	C17

**Table 8 biomolecules-10-00025-t008:** Summary of proton DEPT-135 analysis.

δ-^13^C	No. of Carbon Atoms	Expexted Carbon	Position of Carbon Chain
130.786	1C	-C=C-	C3
128.759	1C	-C=C-	C4
68.141	1C	-CH_2_-0H	C19
38.757	1C	-CH-	C18
30.364	2C	-CH_2_-	C6,C7
29.636	2C	-CH_2_-	C8,C9
28.900	2C	-CH_2_-	C10,C11
23.754	2C	-CH_2_-	C12,C13
22.920	2C	-CH_2_-	C14,C15
13.950	1C	-CH_2_-	C16
10.899	1C	-CH_3_	C17

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
