# Peer review of "Elicitation of Novel Trichogenic-Lipid Nanoemulsion Signaling Resistance Against Pearl Millet Downy Mildew Disease"

_biomolecules, 2019, doi:10.3390/biom10010025_

Round 1

Reviewer 1 Report

In section 3.3. the results were not shown.

About results showed in Table 3, I suggest that it is important to point out that no difference was found  between disease protection mediated by T. B. or  chemical positive control (line 521).

In section 3.5, I consider that HR is a complex molecular event that in most cases is not visually manifested and its establishment only depends of gene-for-gene relationship. I would prefer using the term HR-like lesion.

In Figure 5 you have to explain the meaning of the rows.

In figures 6 and 9 are necessary include the letters of the panels.

Figures 16, 17 and Table 9 can be showed in supplementary data.

In Figure 18 you must to indicate the target gene (LOX, AOC and alpha-DOX) and show with letters or symbols the significant differences among the treatments.

In line 787 as part or your conclussion you have to stand out that T. B. is able to trigger the defense response in a susceptible cultivar in a similar way that a resistant cultivar at least for defense gene expression.

In discussion you can exemplify plant defense triggered by sphingolipid molecules making reference to FB1, that is classified as sphinganine analog mycotoxin produced by Fusarium verticillioides.

Author Response

Responses to the Reviewers’ Comments

The authors would like to thank the Reviewers for their constructive comments and suggestions. We have revised our manuscript according to the Reviewer’s comments. In addition, extensive revision has been undertaken and, all the corrections and suggestions raised by the Reviewer(s) have been incorporated in the revised manuscript.

Reviewers' comments:

Reviewer #1

Response: We would like to express our special thanks to the Reviewer for the positive evaluation of our manuscript, we have taken all our efforts to revise the manuscript, taking into account all the comments and suggestions of the Reviewer that have helped us improve the quality of our manuscript.

Comment 1: In section 3.3. The results were not shown.

Response: Thank you so much for this comment. In the section 3.3, we have the provided the most promisive results of anti-mildew activity and zoosporicidal assay obtained from the positive control metalaxyl fungicide. While the Trichoderma-mediated lipid extracts was not able to inhibit zoospore release and arresting the motility of zoospore. These data was depicted in simple sentences in the result section 3.3(L.510-516).

Comment 2: About results showed in Table 3, I suggest that it is important to point out that no difference was found between disease protection mediated by T. B. or  chemical positive control (line 521).

Response: Thank you so much for this suggestion. Following your advice, we have now incorporated the following sentences in the revised manuscript:

“No significant differences were observed between disease protection mediated by T. brevicompactum and positive control” (L.525-526).

Comment 3: In section 3.5, I consider that HR is a complex molecular event that in most cases is not visually manifested and its establishment only depends of gene-for-gene relationship. I would prefer using the term HR-like lesion.

Response: We thank the Reviewer for this critical observation. As suggested by you we have now replaced HR with HR-like lesion in the revised manuscript (L.560-566).

Comment 4: In Figure 5 you have to explain the meaning of the rows.

Response: We highly appreciate the Reviewer for this comment. Arrows indicate the accumulation of hypersensitive reaction in pearl millet seedlings (L.570-571).

Comment 5: In figures 6 and 9 are necessary include the letters of the panels.

Response: Thank you very much for this comment and observation. With due respect to the Reviewer, we have incorporated the letters in the Fig 6 and Fig 9 in the revised manuscript (P.17 and P.21).

Comment 6: Figures 16, 17 and Table 9 can be showed in supplementary data.

Response: We highly appreciate the Reviewer for this suggestion with which we totally agree. Following your advice, we have now shifted Figures 16, 17 and Table 9 to supplementary data “Supplementary Figure 1, Supplementary 2 and Supplementary Table 1).

Comment 7: In Figure 18 you must to indicate the target gene (LOX, AOC and alpha-DOX) and show with letters or symbols the significant differences among the treatments.

Response: We thank the Reviewer again for this comment. The Figure 18 is now shifted to Figure 16. The significant differences (a, b, c) on the bar graph in the Fig 18 are now mentioned in the revised manuscript (P.28).

Comment 8: In line 787 as part or your conclussion you have to stand out that T. B. is able to trigger the defense response in a susceptible cultivar in a similar way that a resistant cultivar at least for defense gene expression.

Response: Thank you very much for this comment and suggestion. As recommended by you, the sentence is revised as follows:

 It is also observed that T. brevicompactum primed plants are capable of triggering defense responsive genes in the susceptible cultivar in a similar way that of resistant cultivar treated with T. brevicompactum, however the level of gene expression is low in susceptible cultivars.” (L.859-861).

Comment 9: In discussion you can exemplify plant defense triggered by sphingolipid molecules making reference to FB1, that is classified as sphinganine analog mycotoxin produced by Fusariumverticillioides.

Response: The authors would like to thank the Reviewer for this remark. We have now included the appropriate below sentence and reference for this sentence

 Torre-Hernandez et al. [49] illustrated plant defense triggered by sphingolipid molecules i.e., Fumonisin B1 stimulates activation of nuclease and salicylic acid accumulation through sphingoid long-chain base build-up in germinating maize” (L.933-935).

“Torre-Hernandez, M.E. D.; Rivas-San, V.M.;  Greaves-Fernandez, N.; Cruz-Ortega R.;  Plasencia J. Fumonisin B1 induces nuclease activation and salicylic acid accumulation through long-chain sphingoid base build-up in germinating maize. Physiol. Mol. Plant Pathol. 2010, 74, 337-345” (L.1092-1094).

Reviewer 2 Report

These authors need the help of an English-speaking editor. The Materials and methods needs to have sufficient information that someone could repeat the study. For example:

- Line 83: pathogen and inoculum preparation needs to be described

- Line 125 and 131: join both methodologies

- Line 149: … check the effectiveness in controlling the DM disease of PM?, the pathogen was not inoculated in this experiment.

- Line 173: … different concentrations….which?

- line 169-178: how many repetitions were used?

- Line 186: distilled water….sterile?

- Line 219:  zoospore suspension…..zoospore origin?

Author Response

Reviewer #2

We are very glad that the Reviewer highly evaluated our manuscript, and provided constructive comments and suggestions that have helped us improve the quality of our manuscript.

Comment 1: These authors need the help of an English-speaking editor. The Materials and methods needs to have sufficient information that someone could repeat the study. For example:

- Line 83: pathogen and inoculum preparation needs to be described

Response: The authors would like to thank the Reviewer for this important comment. With due respect to the Reviewer’s comment, we have now incorporated briefly the methods for pathogen and inoculums preparation in the revised manuscript as follows:

“Briefly, ten downy mildew-infected young leaves from 7042S cultivar were collected in the evening and washed in running tap water to remove the remnants or old sporangia, later the leaves were cut into small pieces and petri dishes lined with wet double layer of blotter discs and the plates were incubated at 20ºC and 95% RH in the dark for overnight. In the early morning, sporangia produced on the leaves were harvested into sterile distilled water and the spore load was adjusted to 4 x 104 zoospores ml-1 using a hemocytometer, and used as a source of inoculum Nandiniet al. [23]” (L.82-88).

Comment 3: - Line 125 and 131: join both methodologies

Response: Thank you very much for this important suggestion which we totally agree. The methodology of both the experiments is combined together in the revised manuscript (L.131-141).

Comment 4: - Line 149: … check the effectiveness in controlling the DM disease of PM?, the pathogen was not inoculated in this experiment.

Response: We are extremely sorry for adding the above pathogen statement. The sentence is now deleted and revised manuscript (L.154-155).

Comment 5: - Line 173: … different concentrations….which?

Response: We have used different concentrations of Trichoderma-mediated lipids and nanoemulsion (50, 100, 150 µg/ml) for anti-mildew activity. The same is now mentioned in the revised manuscript (L.179-180).

Comment 6: - line 169-178: how many repetitions were used?

Response: The experiment was repeated three times independently (L.174).

Comment 7: - Line 186: distilled water….sterile?

Response: Yes, its sterile distilled water. We have corrected the same in the revised manuscript (L.194).

Comment 8: - Line 219:  zoospore suspension…..zoospore origin?

Response: Zoospore suspension (L.213, 226).